# The antimicrobial peptide defensin cooperates with tumour necrosis factor to drive tumour cell death in *Drosophila*

Jean-Philippe Parvy[1]*, Yachuan Yu[1], Anna Dostalova[2], Shu Kondo[3], Alina Kurjan[4], Philippe Bulet[5], Bruno Lemaître[2], Marcos Vidal[1], Julia B Cordero[1,4]*

[1]CRUK Beatson Institute, Glasgow, United Kingdom; [2]Global Health Institute, School of Life Sciences, Ecole Polytechnique Federale de Lausanne, Lausanne, Switzerland; [3]Invertebrate Genetics Laboratory, Genetic Strains Research Center, National Institute of Genetics, Mishima, Japan; [4]Institute of Cancer Sciences, University of Glasgow, Glasgow, United Kingdom; [5]Institute for Advanced Biosciences, CR University Grenoble Alpes, Inserm U1209, CNRS UMR5309, Immunologie Analytique des Pathologies Chroniques, Grenoble, France

**Abstract** Antimicrobial peptides (AMPs) are small cationic molecules best known as mediators of the innate defence against microbial infection. While in vitro and ex vivo evidence suggest AMPs' capacity to kill cancer cells, in vivo demonstration of an anti-tumour role of endogenous AMPs is lacking. Using a *Drosophila* model of tumourigenesis, we demonstrate a role for the AMP Defensin in the control of tumour progression. Our results reveal that Tumour Necrosis Factor mediates exposure of phosphatidylserine (PS), which makes tumour cells selectively sensitive to the action of Defensin remotely secreted from tracheal and fat tissues. Defensin binds tumour cells in PS-enriched areas, provoking cell death and tumour regression. Altogether, our results provide the first in vivo demonstration for a role of an endogenous AMP as an anti-cancer agent, as well as a mechanism that explains tumour cell sensitivity to the action of AMPs.
DOI: https://doi.org/10.7554/eLife.45061.001

*For correspondence:
j.parvy@beatson.gla.ac.uk (J-PP);
julia.cordero@glasgow.ac.uk (JBC)

## Introduction

Vast amount of evidence demonstrates the key role of systemic immunity in tumour progression and patient outcome. Efforts to manipulate the immune response to tumours are at the core of basic and translational cancer research. In mammals and flies, tumourigenesis triggers inflammation and activation of the immune system, leading to tumour cell death (*Cordero et al., 2010*; *Parameswaran and Patial, 2010*; *Parisi et al., 2014*; *Teng et al., 2008*). Tumour Necrosis Factor (TNF) is an important player in these tumour/immune interactions and has pleiotropic effects on tumours, including induction of cell death (*Ham et al., 2016*; *Parameswaran and Patial, 2010*). This function is conserved in *Drosophila*, where the single TNF homolog Eiger (Egr), produced by tumour-associated macrophages (TAMs), has been shown to drive cell death of neoplastic tumours, generated in larval imaginal discs by the loss of apico-basal complex components such as Disc large 1, Scribble or Lethal giant larvae (*Cordero et al., 2010*; *Parisi et al., 2014*; *Parvy et al., 2018*). Moreover, we have previously reported that tumours from *disc large 1 Drosophila* mutant larvae (*dlg*[40.2] hereafter referred to as *dlg*), activate Toll pathway in the *Drosophila* fat body in an Egr-dependent manner, and this immune activation is necessary for TNF-dependent tumour cell death (*Parisi et al., 2014*). However, the mechanisms by which activation of an immune response in the fat body executes tumour cell death remain unknown (*Figure 1A*).

**eLife digest** Animals have a natural defence system – the immune system – that is needed to fight off disease-causing microbes, known as pathogens. One way the immune system attacks pathogens is by producing small microbe-killing molecules called antimicrobial peptides. These antimicrobial peptides carry a positive charge, which allows them to interact with and disrupt the negatively charged cell surfaces of many microbes. Healthy animal cells do not have these negatively charged components on their cell surface, which means they are invisible to antimicrobial peptides. Studies have reported that antimicrobial peptides can attack cancer cells grown in a dish. However, it was unclear whether they could fight cancer cells in a live animal.

Parvy et al. have now addressed this issue by studying tumours in the larvae of fruit flies. Flies with tumours made an antimicrobial peptide called Defensin, which normally helps to fight infections. When Parvy et al. deleted the gene coding for Defensin, less tumour cells were dying and the tumours became bigger. This result indicated that Defensin was protecting the fruit flies from tumours. Examining the tumours under the microscope showed that Defensin protein interacted with the membranes of tumour cells. Defensin was not, however, interacting with healthy cells.

Further analysis revealed that a negatively charged component of cell membranes called phosphatidylserine, which normally faces the inside of healthy cells, is exposed to the outer surface of tumour cells. This negatively charged molecule renders cancer cells visible to positively charged Defensin. Importantly, the exposure of the phosphatidylserine is mediated by the fly equivalent of a protein called Tumour Necrosis Factor, a key player in cancer biology. Defensin binding to tumour cells leads to their death.

These experiments in the fruit fly highlight key molecular mechanisms that allow antimicrobial peptides to fight cancer cells in a living organism. Because human tumour cells can also expose phosphatidylserine, these latest findings may open up the possibility of a new kind of anti-cancer therapy for patients.

DOI: https://doi.org/10.7554/eLife.45061.002

In *Drosophila* as in mammals, Toll pathway is well known to play a central role in the innate immune response to infection (*Lemaitre et al., 1996*). Downstream effectors of the Toll pathway include antimicrobial peptides (AMPs), which possess microbicidal activities against various pathogens. They display potent antimicrobial activity in vitro by disrupting negatively-charged microbial membranes. While intracellular activities have been reported, many AMPs kill pathogens by inserting into the lipid bilayer and disrupting the membrane integrity (*Brogden, 2005*). Host cells are instead protected from AMP as they are positively charged and contain cholesterol (*Brender et al., 2012*). In vitro studies have revealed AMPs capacity to kill cancer cells (*Deslouches and Di, 2017*). However, whether this cancer-killing activity is a natural function of AMPs is unknown, as there are no reports on an in vivo paradigm addressing such question. Since the Toll pathway is activated in *dlg* mutant tumour bearing larvae and is required for optimal TNF-induced tumour cell death (*Parisi et al., 2014*), we hypothesised that AMPs may be involved in this process.

Here we show that *Drosophila defensin* is induced in the fat body and tracheal system of *dlg* mutant tumour bearing larvae. We find Defensin consistently associated to dying tumour cells. Critically, systemic and tissue specific knockdown of Defensin demonstrates a non-redundant role of the AMP in controlling tumour growth through the induction of tumour cell death. Anti-tumoural Defensin production relies on TNF-dependent activation of both Toll and Imd pathway. Our results demonstrate that *dlg* mutant tumours expose PS in response to haemocyte-derived TNF and that Defensin is present in PS enriched area on the tumour surface. Finally, we find that lack of TNF prevents PS exposure in tumours and makes them insensitive to the action of Defensin. Collectively, our results reveal an anti-tumoural role for Defensin in vivo and provide insights into the molecular mechanisms, which make tumours sensitive to the killing action of an endogenous AMP.

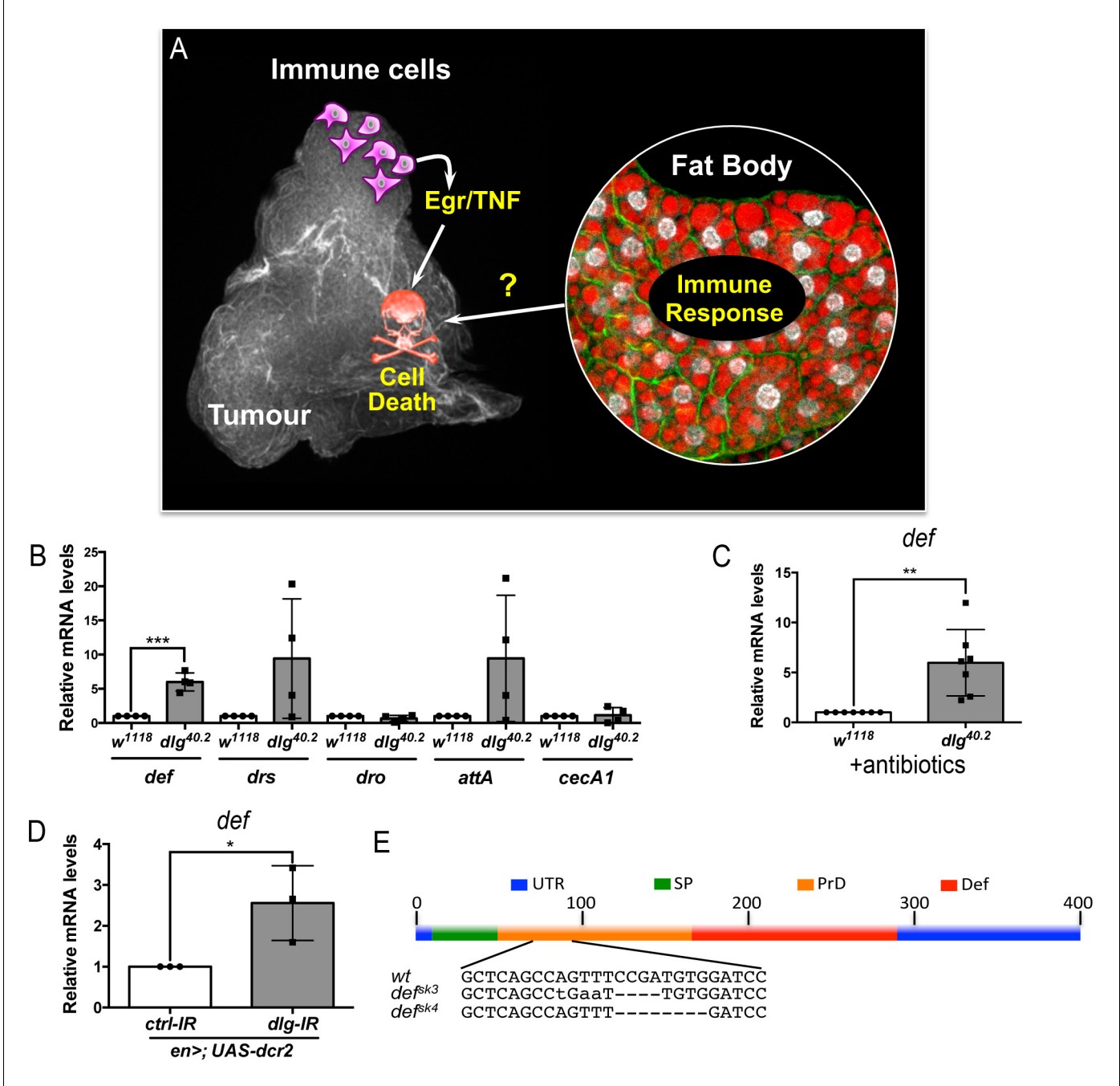

**Figure 1.** *def* is induced in *dlg* mutant tumour bearing animals. (**A**) Working model showing the cooperation between haemocyte-derived TNF and the immune response in the fat body in tumour cell death (*Parisi et al., 2014*). (**B**) RT-qPCR analyses showing expression of several AMPs in the fat body of *dlg$^{40.2}$* mutant tumour bearing larvae compared to wild-type (*w$^{1118}$*) larvae (n = 4). (**C**) RT-qPCR analysis of *def* expression in *w$^{1118}$* and *dlg$^{40.2}$* whole larvae reared on antibiotics (n = 7). (**D**) RT-qPCR analysis showing *def* expression in larvae expressing a *ctrl-IR* or a *dlg-IR* in the posterior part of the wing disc (*en>;UAS-dcr2*) (n = 3). (**E**) Schematic representation of the *def* gene locus showing mutant alleles generated (UTR: Untranslated Regions, SP: Signal Peptide, PrD: Pro-Domain, Def: Mature Defensin). Statistical analysis: B-D, Student t-test, B, ***p=0.0003, C, **p=0.0074, D, *p=0.042.

DOI: https://doi.org/10.7554/eLife.45061.003

The following figure supplement is available for figure 1:

**Figure supplement 1.** *def* mediates animal survival to infection by Gram-positive bacteria.

DOI: https://doi.org/10.7554/eLife.45061.004

## Results

### *dlg* tumour bearing larvae express the AMP defensin

In order to assess expression of several AMPs we performed RT-qPCR analysis on fat bodies dissected from wild type controls ($w^{1118}$) or *dlg* mutant larvae (*Figure 1B*). Results showed consistent and statistically significant upregulation of *defensin* in fat bodies of *dlg* larvae compared to wild-type ones (*Figure 1B*). Other Toll-dependent AMPs display a trend to be increased (*drosomycin* and *attacin A*), even though data were highly variable amongst biological replicates, while other AMPs were not transcriptionally regulated (*drosocyn*, *cecropin A1*) (*Figure 1B*). Interestingly, human β-Defensin-1 displays anticancer activity in vitro (*Bullard et al., 2008*; *Sun et al., 2006*), and deletion of human *def* appears prevalent in some cancer types (*Ye et al., 2018*). This prompted us to explore the role of *Drosophila* Defensin in *dlg* mutant tumours. Using *dlg* larvae reared on antibiotics, we confirmed that *defensin* upregulation was independent of the presence of microbes (*Figure 1C*). Moreover, larvae bearing *dlg* imaginal discs tumours induced by RNAi (*en >UAS-dcr2; UAS-dlg-IR*) also displayed increased *defensin* expression, confirming *def* gene induction as a consequence of *dlg*-driven epithelial transformation rather than whole body dl loss (*Figure 1D*). We conclude that Defensin, an AMP known for its activity against microbes, is induced, in a sterile environment, by the presence of tumours.

### Defensin restrains *dlg* tumour growth and promotes tumour cell death

We next hypothesised that Defensin may be an important mediator of anti-tumour immunity in vivo. To test this hypothesis, we generated null mutant alleles for the *defensin (def)* gene using the CRISPR/Cas9 system (*def^{sk3}* and *def^{sk4}*) (*Figure 1E*) (*Hanson et al., 2019*). Survival analysis of *def^{sk3}* flies confirmed that this AMP contribute to resist systemic infection to certain Gram-positive bacteria (*Figure 1—figure supplement 1*) (*Levashina et al., 1995*). To evaluate the effect of Defensin on tumour development, we combined *def* and *dlg* loss of function alleles. Compared to *dlg* mutant animals, *dlg;def* double mutants displayed a significant increase in tumour size (*Figure 2A*). Tumour growth is limited by apoptotic tumour cell death as revealed by Dcp1 staining (*Parisi et al., 2014*). Interestingly, tumours from *dlg;def* double mutants display a very strong decrease in apoptosis (*Figure 2B–E′*), suggesting that increased tumour size in absence of Defensin is due to a decrease in tumour cell death. This was further supported by the similar proliferation rates measured in *dlg* and *dlg,def* mutant tumours as per quantification of anti-phophoHistone H3 staining (*Figure 2—figure supplement 1A–E′*). Importantly, the effect of Def on tumour size and cell death were still observed when *dlg* and *dlg,def* larvae were reared in sterile conditions (*Figure 2—figure supplement 1F–J′*) and further confirmed upon ubiquitous knock down of *def* using RNA interference (IR) (*Figure 2F–J′*). Furthermore, fat body overexpression of *def* significantly rescued tumour volume and tumour cell death of *dlg;def^{sk3}* double mutant animals (*Figure 2K and L*). Additionally, larval injection of a synthetic Defensin peptide increased tumour cell death of *dlg* or *dlg;def^{sk3}* imaginal discs (*Figure 2M*), while it had no effect on tissues from wild-type larvae, indicating that Defensin can selectively promote cell death of tumour cells. Altogether, these results demonstrate that Defensin is required to control *dlg*-dependent tumourigenesis in vivo through induction of tumour cell death.

### Defensin remotely produced from immune tissues bind to tumour cells

Having shown that Defensin restrict tumour growth, we sought to determine the tissues that produced endogenous Defensin in the context of tumour bearing. Previous studies have shown that Defensin is not produced by imaginal discs or tumours (*Bunker et al., 2015*; *Külshammer et al., 2015*). In the context of infection, Defensin can be produced by the fat body as well the tracheal and gut epithelium (*Tzou et al., 2000*). We monitored *defensin* expression by RT-qPCR in various immune tissues of tumour bearing animals. We observed that the fat body, homologue to the mammalian liver and adipose tissue, and the trachea, a network of tubes transporting oxygen to cells that resembles the mammalian vasculature, were the main sources of *defensin* in these animals (*Figure 3A*). This was also supported by Defensin immunostaining, (*Figure 3B and C*). Transcript assessment upon targeted IR knock down of *defensin* in the respective tissues further confirmed the domains of endogenous gene expression (*Figure 3D and I*). Importantly, knocking down *defensin* expression specifically in the fat body or the trachea of *dlg* animals, resulted in increased tumour

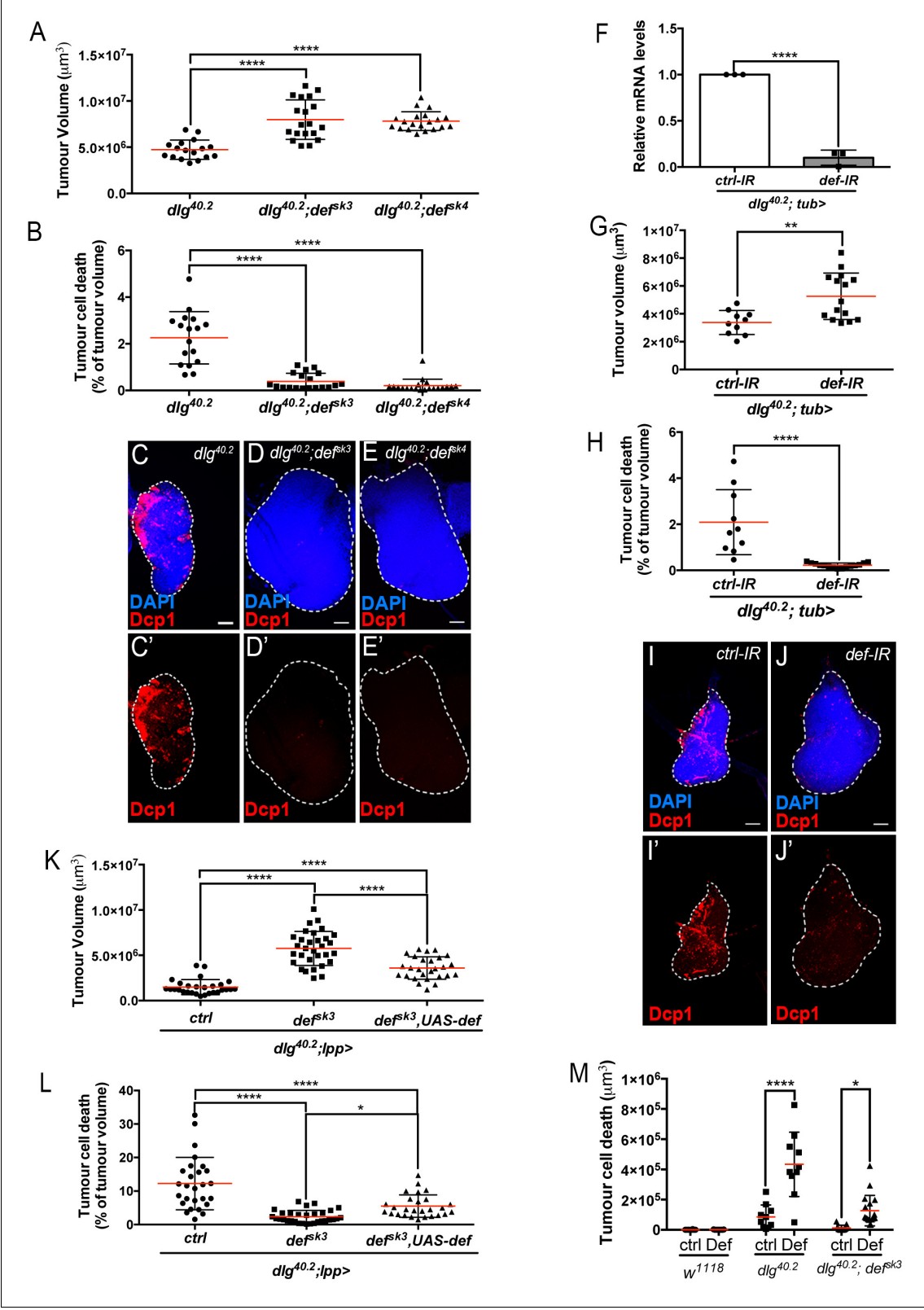

**Figure 2.** Def restricts tumour growth and promotes tumour cell death. (A-E') Quantification of tumour volume (TV) (A) and tumour cell death (TCD) (B) in wing imaginal discs from $dlg^{40.2}$ (n = 17), $dlg^{40.2};def^{sk3}$ (n = 19) and $dlg^{40.2};def^{sk4}$ (n = 20) mutant larvae and representative immunofluorescence images of tissues quantified (C–E'). F, RT-qPCR analysis showing $def$ expression upon ubiquitous knockdown ($dlg^{40.2};$ $tub>$) (n = 3). G-J', Quantification of TV (G) and TCD (H) in wing imaginal discs from larvae ubiquitously expressing a $control-IR$ ($ctrl-IR$: n = 10) or a $def-IR$ (n = 15) and representative

*Figure 2 continued on next page*

Figure 2 continued

immunofluorescence images of tissues quantified (I–J'). K-L, Quantification of TV (K) and TCD (L) in wing imaginal discs from *dlg* mutant controls (*dlg*$^{40.2}$;*lpp*>: n = 28), *dlg;def*$^{sk3}$ mutants controls (*dlg*$^{40.2}$;*def*$^{sk3}$;*lpp*>: n = 31) or *dlg;def*$^{sk3}$ mutants expressing *def* in the fat body (*dlg*$^{40.2}$;*def*$^{sk3}$,*UAS-def*; *lpp*>: n = 27). M, Effect of PBS (ctrl) or synthetic Def injection on TCD from wild-type larvae (*w*$^{1118}$, ctrl: n = 10, Def: n = 9), *dlg*$^{40.2}$ (ctrl: n = 10, Def: n = 10) and *dlg*$^{40.2}$;*def*$^{sk3}$ (ctrl: n = 11, Def: n = 18) mutant larvae. Tissues were stained with 4',6-diamidino-2-phenylindole (DAPI, blue) for nuclei visualisation and with anti-cleaved Decapping protein 1 (Dcp1) antibody (red) to detect apoptotic cell death. Scale bars = 50 μm. Statistical analysis: A, B, K, L, One way ANOVA, *p<0.05, ****p<0.0001; F, Student t-test, ****p<0.0001, G-H, Mann-Whitney test, G, **p=0.0054, H, ****p<0.0001; M, Two way ANOVA (only relevant significant statistics are indicated), *p<0.05, ****p<0.0001.
DOI: https://doi.org/10.7554/eLife.45061.005
The following figure supplement is available for figure 2:

**Figure supplement 1.** Tumour suppression by Def is independent of infection and tumour cell proliferation.
DOI: https://doi.org/10.7554/eLife.45061.006

size and decreased tumour cell death (*Figure 3E–H' and J–M'*) confirming the non-redundant functional requirement of *defensin* in both tissues to efficiently promote tumour cell death.

To investigate the possibility that Defensin produced by the fat body and the trachea can specifically target tumour cells, we made use of an inducible haemagglutinin (HA) tagged form of Defensin (*UAS-def-HA*). We noticed leaky HA protein expression in the tracheal system (*Figure 4—figure supplement 1A*) but not in the fat body (*Figure 4—figure supplement 1B*) of animals carrying the *UAS-def-HA* construct only. Therefore, we overexpressed *UAS-def-HA* in the fat body using a Gal4 specific driver (*lpp >def* HA), which resulted in significant upregulation of Defensin in the fat body (*Figure 4—figure supplement 1D*) when compared to control conditions (*Figure 4—figure supplement 1B*), but also maintained the leaky expression of the transgene in the trachea (*Figure 4—figure supplement 1C*). Strikingly, overexpression of *UAS-def-HA* in the fat body resulted in Defensin-HA immunostaining in transformed imaginal discs from *dlg* mutant animals (*Figure 4A–A'' and B–B''*) but not in normal tissues from *dlg* heterozygous animals (*Figure 4C and C'*).

Consistently, using anti-Defensin antibody, we observed endogenous Defensin staining on *dlg* mutant tumour (*Figure 4D–D'' and E–E''*) but not in wild-type discs (*Figure 4G,G'*). Interestingly, we observed Defensin preferentially bound to tumour areas enriched with apoptotic cells (*Figure 4D–D'' and E–E''*). This was confirmed by quantification of the amount of Def staining colocalising with Dcp1 staining (*Figure 4F*). High-resolution imaging showed Defensin enrichment at the membrane of these dying cells (*Figure 4H–H''*).

Altogether, these results show that Defensin produced by immune tissues selectively binds tumour cells to target them for apoptosis.

## Toll and Imd pathway contribute to Defensin expression in *dlg* mutant larvae

*defensin* expression upon systemic infection relies on both Toll and Imd pathways (*Lemaitre et al., 1996*; *Leulier et al., 2000*). While we previously showed that Toll pathway activation in *dlg* mutant larvae is required to achieve maximal tumour cell death (*Parisi et al., 2014*), the involvement of the Imd pathway in tumour bearing animals was still elusive. To assess the contribution of the Imd pathway to both *defensin* expression and tumour burden, we analysed these two phenotypes in larvae deficient for the Imd-pathway. We observed a 55–60% decrease in *defensin* expression in *dlg* mutants carrying a loss of function allele affecting *imd* (*dlg;imd*$^1$) (*Figure 5A*) or the gene encoding the downstream transcription factor Relish (*dlg;rel*$^{E20}$) (*Figure 5F*). Consistently, analysis of tumour phenotypes revealed increased tumour volume and decreased tumour cell death in *dlg;imd*$^1$ and *dlg;rel*$^{E20}$ animals when compared with *dlg* counterparts (*Figure 5B–E' and G–J'*). Altogether, these data demonstrate that the Imd pathway is required for *defensin* upregulation, impairment of tumour growth and induction of tumour cell death in *dlg* mutant larvae.

Upon infection, AMP expression in the trachea exclusively relies on the Imd pathway (*Tzou et al., 2000*). We therefore looked at *defensin* expression and tumour phenotype in *dlg* mutant larvae where *imd* expression had been knocked down specifically within tracheal cells (*dlg;btl >imd-IR*). In this setting, *defensin* expression was significantly reduced in the whole larvae (*Figure 6A*). Consistently, tumour volume was increased while tumour cell death was decreased (*Figure 6B–E'*) showing the requirement of Imd pathway in the tracheal system to control tumour burden.

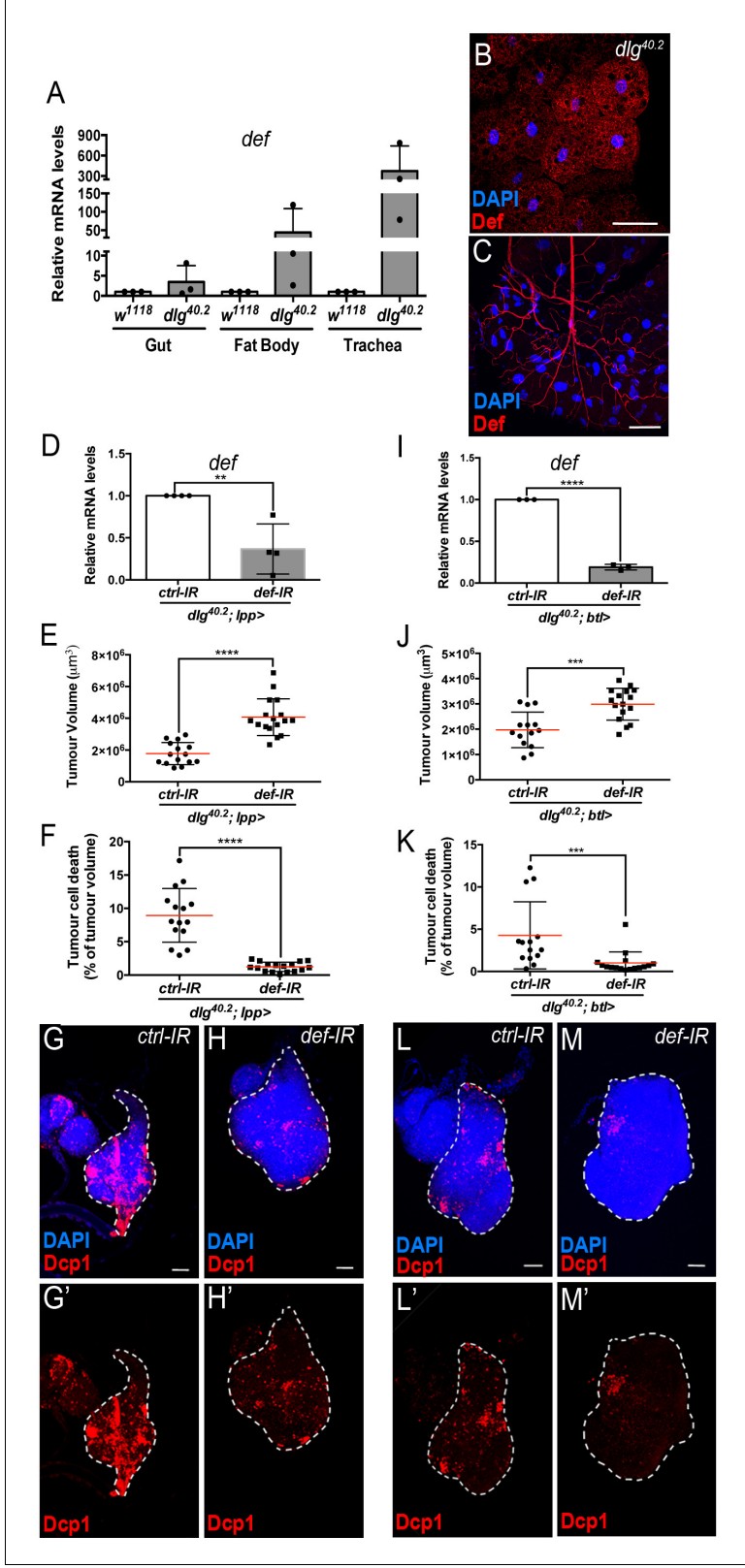

**Figure 3.** Def from the trachea and the fat body mediates tumour cell death. (**A**) RT-qPCR analysis of *def* expression in gut, fat body and trachea dissected from *w^1118* or *dlg^40.2* mutant larvae (n = 3). **B-C**, Fat body (**B**) and trachea (**C**) from *dlg^40.2* mutant larvae stained with DAPI (blue) and anti-Def antibody (red). **D**, RT-qPCR analysis of *def* expression in *dlg^40.2* mutant larvae (*dlg^40.2;lpp>*) expressing a *ctrl-IR* or *def-IR* in the fat body (n = 4). **E-H'**, *Figure 3 continued on next page*

*Figure 3 continued*

Quantification of TV (E) and TCD (F) in wing imaginal discs from $dlg^{40.2}$ mutant larvae ($dlg^{40.2}$;*lpp>*) expressing a *ctrl-IR* (n = 15) or *def-IR* (n = 17) in the fat body and representative immunofluorescence images of tissues quantified (G–H'). I, RT-qPCR analysis of *def* expression in dissected trachea from $dlg^{40.2}$ mutant larvae ($dlg^{40.2}$; *btl>*) expressing a *ctrl-IR* or *def-IR* in the trachea (n = 3). J-M', Quantification of TV (J) and TCD (K) in wing imaginal discs from $dlg^{40.2}$ mutant larvae ($dlg^{40.2}$;*btl>*) expressing a *ctrl-IR* (n = 14) or *def-IR* (n = 16) in the trachea and representative immunofluorescence images of tissues quantified (L–M'). Tumours were stained with DAPI (blue) and anti-Dcp1 antibody (red). Scale bars = 50 µm. Statistical analysis: A, D, I, Student t-test, D, **p=0.0054, I, ****p<0.0001; E, F, J, K, Mann-Whitney test, E, F, ****p<0.0001, J, ***p=0.0009, K, ***p=0.0004.

DOI: https://doi.org/10.7554/eLife.45061.007

In the fat body, both Imd and Toll pathways contribute to AMPs expression during infection (*Tzou et al., 2002*). To evaluate the contribution of Toll and Imd pathways in fat body in tumour bearing larvae, we monitored *defensin* expression and tumour phenotype in *dlg* animals in which Toll or Imd pathways have been selectively knocked down in the fat body. Knocking down *myd88,* which encodes a common adaptor to Toll receptors (*Imler and Zheng, 2004*), or *imd* in the fat body resulted in a significant decrease in *defensin* expression (*Figure 6F*) and, consistently, increased tumour volume and decreased tumour cell death (*Figure 6G–K'*). Moreover, concomitant overexpression of *defensin* and *Myd88-IR* or *Imd-IR* in the fat body of *dlg* larvae was sufficient to significantly rescue the tumour volume and tumour cell death phenotypes resulting from fat body knockdown of *Myd88* or *Imd* in *dlg* animals (*Figure 6L–O*).

We conclude that, in *dlg* mutant animals, Toll and Imd pathways have non-redundant roles in restricting tumour growth and promoting tumour cell death through the control of *defensin* expression. Importantly, forced *defensin* expression in an otherwise immune-compromised animal is sufficient to reduce tumour growth and to promote tumour cell death.

## Defensin is enriched in tumour areas exposing phosphatidylserine

AMPs targeting of pathogens, involves the recognition of negatively charged molecules exposed on the cell surface (*Yeaman and Yount, 2003*). A key question raised by our study is how Defensin can selectively bind and kill tumour cells (*Figure 4* and *Figure 2M*). Selective action of cationic AMP is attributed to their ability to interact with negatively charged membrane such as those found in bacteria. We hypothesised that the membrane of tumour cells from *dlg* mutant larvae might change their electrostatic properties, making them sensitive to the action of Defensin. Phosphatidylserine (PS) is a negatively charged phospholipid, normally restricted to the inner leaflet of the cell membrane. However, PS can be exposed on the outer leaflet for example in apoptotic cells, which tags these cells for phagocytosis (*Birge et al., 2016*; *Shklyar et al., 2013*; *Tung et al., 2013*). Moreover, PS exposure has been shown to occur independently of apoptosis in many cancer cell types (*Riedl et al., 2011*). Therefore, we investigated whether PS externalisation could be a factor allowing specific targeting of tumour cells by Defensin. Our data revealed that *dlg* but not wild-type tissues, displayed high levels of Annexin V staining (*Figure 7A and F*; compare to E), indicating increased exposure of PS by *dlg* tumours. We next looked at the ability of Defensin to specifically associate with tumour cells exposing PS and found that Defensin was enriched in Annexin V$^{+ve}$ areas on *dlg* tumours (*Figure 7A–A'' and B–B''*). This was confirmed by quantification of the amount of Def staining colocalising with Annexin V staining (*Figure 7C*). Thus, Defensin produced by immune responsive tissues binds specifically to tumour cells and this ability correlated with their exposure of PS.

## The TNF homolog Eiger is required for PS exposure and defensin antitumoural activity

Previous studies have shown that circulating macrophage-like cells in *Drosophila,* called haemocytes, bind to *dlg* mutant tumours and contribute to cell death (*Parisi et al., 2014*). This process is mediated by the release of Egr from haemocytes (*Cordero et al., 2010*; *Parisi et al., 2014*), which then activates the JNK pathway in target cells to promote apoptosis (*Igaki et al., 2009*). Moreover, Toll pathway activation in tumour bearing animals also requires Egr (*Parisi et al., 2014*). Accordingly, we found that *defensin* upregulation observed in *dlg* animals was lost in *dlg;egr*[3] double mutants (*Figure 7D*). We next tested whether PS exposure in tumours was dependent on Eiger, and

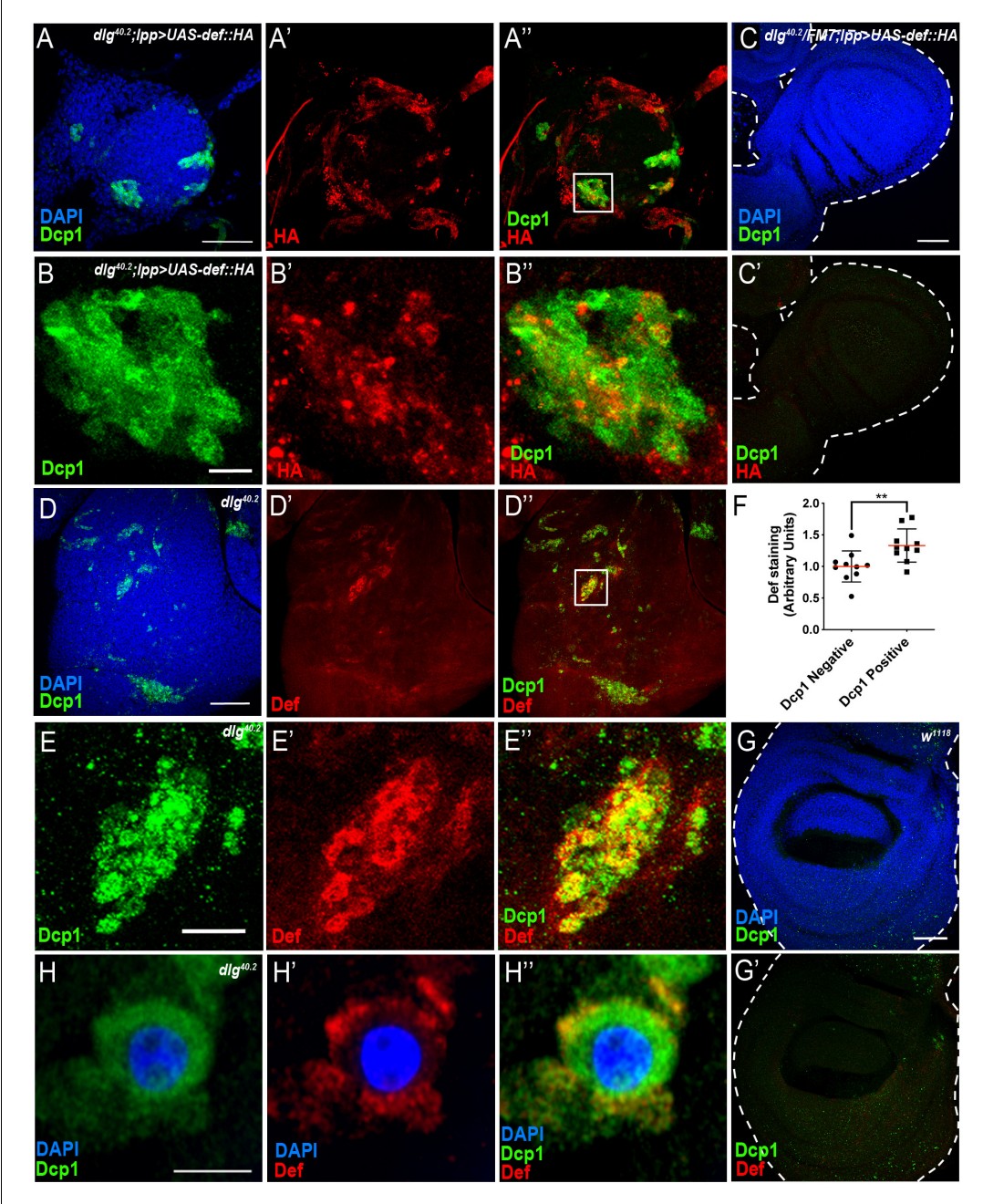

**Figure 4.** Def produced by immune tissues specifically targets tumour cells. (**A-A''**) DAPI (blue), anti-Dcp1 (green) and anti-HA antibody (red) staining of $dlg^{40.2}$ mutant tumour from larvae overexpressing a *def-HA* construct in the fat body and the trachea ($dlg^{40.2};lpp > UAS-def-HA$). **B-B''**, Enlargement of inset from **A''** (white outline) showing Dcp1 (**B**), Def-HA (**B'**) and merged channels (**B''**). **C-C'**, DAPI (blue), anti-Dcp1 (green) and anti-HA antibody (red) staining of $dlg^{40.2}$ heterozygous wing disc from larvae overexpressing *def-HA* ($dlg^{40.2}/FM7;lpp > UAS-def-HA$). **D-D''**, $dlg^{40.2}$ mutant tumour stained with DAPI (blue), anti-Def (red) and anti-Dcp1 (green) antibodies. **E-E''**, Enlargement of inset from **D''** (white outline) showing Dcp1 (**E**), Def (**E'**) and merged channels (**E''**). F, Quantification of colocalization between Def and Dcp1 staining (n = 10). **G-G'**, wild type ($w^{1118}$) wing imaginal disc stained with DAPI (blue), anti-Def (red) and anti-Dcp1 (green) antibodies. **H-H''**, High-resolution imaging of a single dying tumour cell stained with DAPI (blue), anti-Def (red) and anti-Dcp1 (green) antibodies. A, C, D, G, Scale bars = 50 μm; B, E, Scale bars = 10 μm; H, Scale bar = 2.5 μm. Statistical analysis: F, Student t-test, **p=0.0093.

DOI: https://doi.org/10.7554/eLife.45061.008

The following figure supplement is available for figure 4:

**Figure supplement 1.** Characterisation of UAS-def-HA expression.
DOI: https://doi.org/10.7554/eLife.45061.009

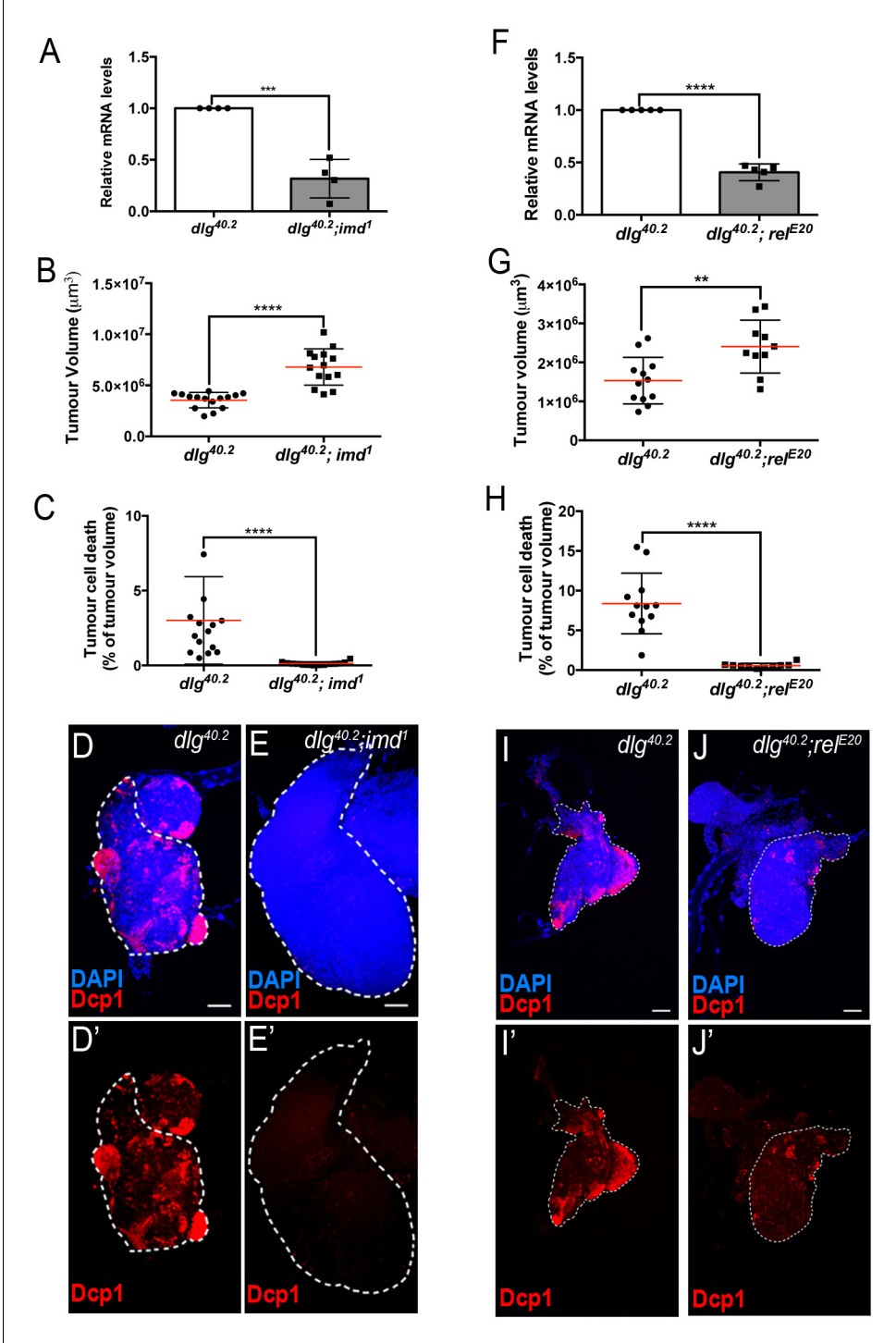

**Figure 5.** Imd pathway activation is required for *def* expression and Def-mediated tumour cell death. (A) RT-qPCR analysis of *def* expression from *dlg[40.2]* and *dlg[40.2];imd[1]* mutant larvae (n = 4). B-E', Quantification of TV (B) and TCD (C) in wing imaginal discs from *dlg[40.2]* (n = 15) and *dlg[40.2];imd[1]* (n = 14) mutants larvae and representative immunofluorescence images of tissues quantified stained with DAPI (blue) and anti-Dcp1 antibody (red) (D–E'). F, RT-qPCR analysis showing *def* expression in *dlg[40.2]* and *dlg[40.2]; rel[E20]* mutant animals (n = 5). G-J', Quantification of TV (G) and TCD (H) from *dlg[40.2]* (n = 12) and *dlg[40.2];rel[E20]* (n = 10) mutant larvae and representative pictures of the corresponding tumours (I–J'). Scale bars = 50 μm. Statistical analysis: A, F, Student t-test, A, ***p<0.0003, F, ****p<0.0001; B, C, G, H, Mann-Whitney test, B, C, H, ****p<0.0001, G, **p=0.009.
DOI: https://doi.org/10.7554/eLife.45061.010

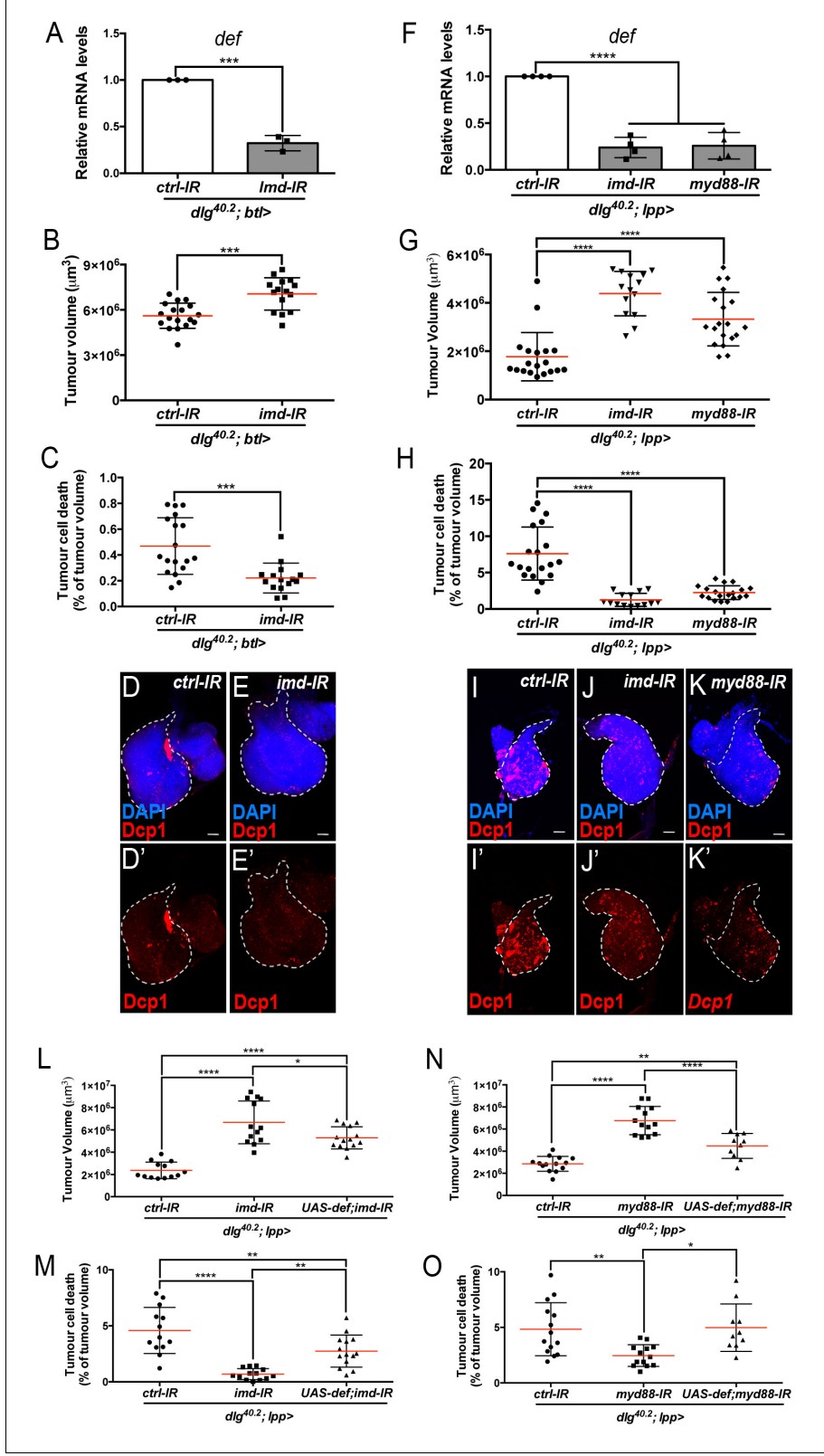

**Figure 6.** Imd and Toll pathway activation are required in the trachea and the fat body to promote Defensin-dependent tumour cell death. (**A**) RT-qPCR analysis of *def* expression from *dlg^{40.2}* mutant larvae (*dlg^{40.2};btl>*) expressing a *ctrl-IR* or *imd-IR* in the trachea (n = 3). **B-E'**, Quantification of TV (**B**) and TCD (**C**) from *dlg^{40.2}* mutant larvae (*dlg^{40.2};btl>*) expressing a *ctrl-IR* (n = 18) or *imd-IR* (n = 15) in the trachea and representative

*Figure 6 continued on next page*

*Figure 6 continued*
immunofluorescence images of tissues quantified stained with DAPI (blue) and anti-Dcp1 antibody (red) (D–E'). F, RT-qPCR analysis of *def* expression from *dlg*$^{40.2}$ mutant larvae (*dlg*$^{40.2}$;*lpp>*) expressing a *ctrl-IR*, an *imd-IR* or a *myd88-IR* in the fat body. G-K', Quantification of TV (G) and TCD (H) from *dlg*$^{40.2}$ mutant larvae (*dlg*$^{40.2}$;*lpp>*) expressing a *ctrl-IR*, an *imd-IR* or a myd88-IR in the fat body and representative immunofluorescence images of tissues quantified (I–K'). L-O, Quantification of TV (L, N) and TCD (M, O) from *dlg*$^{40.2}$ mutant larvae (*dlg*$^{40.2}$;*lpp>*) expressing a *ctrl-IR*, an *imd-IR* alone or in combination with a *UAS-def* in the fat body (L, M) and from *dlg*$^{40.2}$ mutant larvae (*dlg*$^{40.2}$;*lpp>*) expressing a *ctrl-IR*, a *myd88-IR* alone or in combination with a *UAS-def* in the fat body (N, O). Scale bars = 50 μm. Statistical analysis: A, Student t-test, A, ***p=0.0001; B, C, Mann-Whitney test, B, ***p=0.0003, C, ***p=0.0002; F, G, H, L-O, One way ANOVA, *p<0.05, **p<0.01, ****p<0.0001.
DOI: https://doi.org/10.7554/eLife.45061.011

observed an almost complete loss of cell-surface PS in *dlg;egr*$^3$ tumours (*Figure 7E–G*). Therefore, Egr is required for PS exposure in these tumours. We then analysed PS exposure in *dlg* mutant tumours, upon specific *egr* knockdown in haemocytes (*dlg;hml >egr-IR*), and observed a strong decrease in cell surface-exposed PS compared to control tumours (*dlg;hml >ctrl-IR*) (*Figure 7H–J*). This indicates that PS exposure in tumour cells is likely triggered by the release of Egr from haemocytes. Finally, to further test the hypothesis that Def requires PS to bind to tumours, we assessed the ability of overexpressed Def-HA to bind to *dlg,egr*$^3$ mutant tumours. Consistently with the observed basal and overexpressed Def-HA expression pattern in wild type tissues (*Figure 4—figure supplement 1*), Def-HA was expressed in the trachea and fat body of *dlg,egr*$^3$ mutant animals (*Figure 7—figure supplement 1A,B*). However, we did not find any detectable association of Def-HA to the surface of *dlg,egr*$^3$ tumours (*Figure 7—figure supplement 1C,C'*).

Importantly, Egr-dependent signalling and PS exposure were intact in *dlg;def* and *dlg;imd* mutants (*Figure 7—figure supplement 2*) indicating that both events precede the action of Defensin in *dlg* tumours. To further assess whether Egr was required for Defensin-induced tumour cell death, we injected synthetic Defensin peptide into control, *dlg* or *dlg;egr*$^3$ mutant larvae. While, Defensin injection was able to robustly promote tumour cell death in *dlg* mutant tumours, it was unable to affect tumours derived from Egr-deficient animals, further demonstrating the requirement of Egr for tumour cell death induced by Defensin (*Figure 7K*).

Finally, we tested whether the antitumoral action of Def extended to other tumour models. We observed that tumours induced by the loss of *scribble* (*scrib*), another member of the same apico-basal complex as *dlg,* showed significant cell surface PS exposure, which was Egr dependent (*Figure 7—figure supplement 3A,A'*; compare to *Figure 7—figure supplement 3B,B'*). Moreover, we detected Def enrichment in areas positive for Dcp1 staining on *scrib* tumours (*Figure 7—figure supplement 3C–C'' and D–D''*). Consistently, removing *def* from *scrib* mutant animals led to increase in tumour volume and decrease in tumour cell death (*Figure 7—figure supplement 3E–H'*). Altogether, these results show that the sensitivity to Def is not restricted to *dlg* mutant tumours but might rather be a general feature of neoplastic growth induced by loss of cell polarity.

## Discussion

While the role of antimicrobial peptides in innate immune defense has been well-recognised for decades (*Bahar and Ren, 2013*), recent reports revealed potential additional physiological functions of AMPs, including ageing and neurodegeneration (*Cao et al., 2014*; *Kounatidis et al., 2017*; *Lezi et al., 2018*), wound-healing (*Chung et al., 2017*; *Tokumaru et al., 2005*), resistance to oxidative stress (*Mergaert et al., 2017*; *Zhao et al., 2011*), immune signaling (*Tjabringa et al., 2003*; *van Wetering et al., 2002*) and anti-cancer activity (*Deslouches and Di, 2017*). However, due to the absence of AMP mutants, most of these studies have relied on the use of exogenous sources of AMPs or genetic modification of upstream regulators of AMP expression. Recently, the use of loss of function alleles of *Drosophila* Diptericin (DptB), allowed to establish a role for the AMP in long-term memory (*Barajas-Azpeleta et al., 2018*). Pioneer work reporting systematic deletion of multiple AMPs in *Drosophila*, opens the door for in-depth analysis of the endogenous functions of these molecules (*Hanson et al., 2019*). Using such tools and a genetically defined in vivo tumour model, our study demonstrates a role for Defensin in the control of tumours and deciphers the molecular

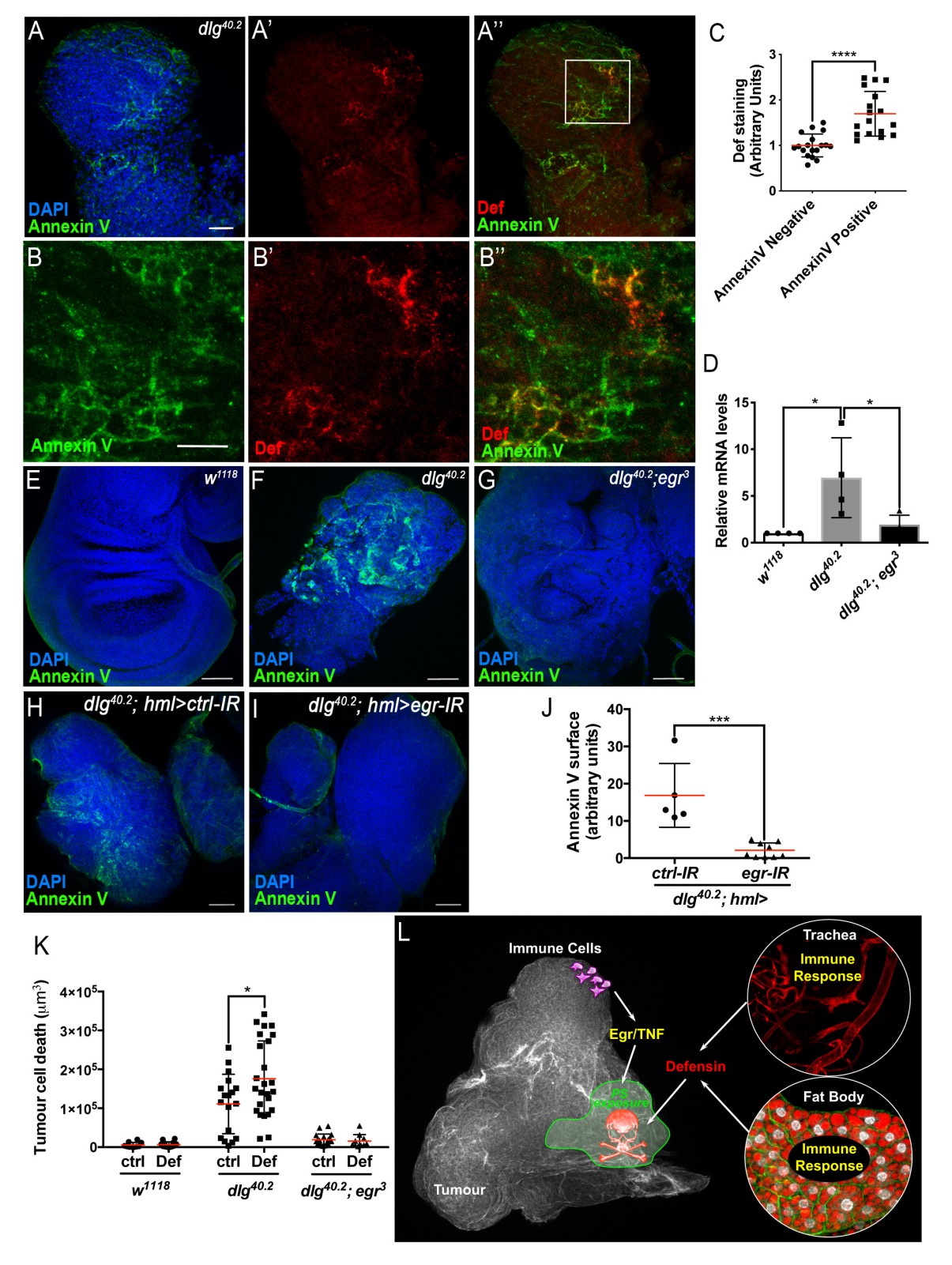

**Figure 7.** TNF is required for PS exposure and Defensin-driven tumour cell death. (**A-A''**) DAPI (blue), Annexin V (green) and anti-Def (red) staining of *dlg^{40.2}* mutant tumours. (**B-B''**) Enlargement of inset from **A''** (white outline) showing Annexin V (**B**), Def (**B'**) and merged channels (**B''**). (**C**) Quantification of colocalisation between Def and Annexin V staining (n = 18). (**D**) RT-qPCR analysis showing *def* expression in wild-type, *dlg^{40.2}* or *dlg^{40.2};egr^3* mutants. (**E- G**) Annexin V (green) and DAPI (blue) staining of wing imaginal discs from larvae of the indicated genotypes. (**H, I**) Annexin V

*Figure 7 continued on next page*

*Figure 7 continued*

(green) and DAPI (blue) staining of wing imaginal discs from *dlg* mutant larvae (*dlg*$^{40.2}$,*hml>*) expressing a *ctrl-IR* (n = 5) or an *egr-IR* (n = 9) in the haemocytes. (J) Quantification of Annexin V signal on tumours from the corresponding genotypes. (K) Quantification of TCD upon PBS (ctrl) or Def injection in wild-type (*w*$^{1118}$, ctrl: n = 10, Def: n = 14), *dlg*$^{40.2}$ (ctrl/Def: n = 18) or *dlg*$^{40.2}$;*egr*$^3$ (ctrl: n = 20, Def: n = 13) mutant larvae. (L) A model for Def antitumoural activity. A, E-I, Scale bars = 50 µm; B, Scale bars = 20 µm. Statistical analysis: C, Student t-test, ****p<0.0001; D, One way ANOVA, *p<0.05; J, Mann-Whitney test, ***p=0.001; K, Two way ANOVA (only relevant significant statistics are indicated), *p<0.05.

DOI: https://doi.org/10.7554/eLife.45061.012

The following figure supplements are available for figure 7:

**Figure supplement 1.** Def-HA does not associate to PS-negative *egr*-mutant tumours.

DOI: https://doi.org/10.7554/eLife.45061.013

**Figure supplement 2.** TNF signalling activation does not require functional *def*.

DOI: https://doi.org/10.7554/eLife.45061.014

**Figure supplement 3.** *scrib* mutant tumours expose PS in a TNF-dependent manner and are sensitive to Def action.

DOI: https://doi.org/10.7554/eLife.45061.015

mechanism allowing specific tumour cell targeting by the AMP. Collectively, our results suggest that the two branches of *Drosophila* innate immunity contribute to tumour elimination (*Figure 7L*). The cellular immune response to tumours involves the binding of haemocytes to tumour cells and activation of TNF pathway leading to PS exposure (*Figure 7L*). This is followed by a humoral response in distant tissues (fat body and trachea) that triggers tumour cell death through the action of Defensin (*Figure 7L*).

## A role for a *Drosophila* AMP in tumour control

Using mutants as well as ubiquitous or targeted knockdowns, our study reveals that Defensin is non-redundantly required to drive tumour cell death and restrict tumour growth in neoplastic tumours generated by loss the apico-basal determinant Dlg and Scrib. This was reinforced by detection of Defensin on dying tumour cells as well as genetic rescue and injection experiments showing that Defensin can actively drive tumour cell death. Several in vitro studies in mammals have pointed to AMPs anti-tumoural potential (*Deslouches and Di, 2017*). Amongst these suggested anticancer peptides, Human β-Defensin-1 (hBD1) appears downregulated in 82% of prostate cancer and 90% of renal clear cell carcinomas (*Donald et al., 2003*). Furthermore, expression of hBD1 induces cell death in prostate and renal cancer cells in vitro (*Bullard et al., 2008*; *Sun et al., 2006*). Moreover, a recent report shows prevalent deletion of Human *defensin* gene cluster in some tumour types including prostate, lung and colorectal cancers, as well as a decrease overall survival in patients carrying these deletions (*Ye et al., 2018*). However, since the classification of a molecule as AMP is based on structural protein features, it is important to mention that Human Defensins are not homologous of, but rather have structural resemblances, to *Drosophila* AMPs due to convergent evolution (*Shafee et al., 2017*). Further studies are required to determine if anticancer activity displayed by Defensins from different species is linked to their structural properties.

As upon infection (*Tzou et al., 2000*), we show that the fat body and the trachea are the main sources of Defensin in tumour bearing larvae. Moreover, our results indicated that Defensin's maximal expression and anti-tumour properties rely on Imd and Toll pathways activation. As expected from previous studies on anti-pathogenic immunity, Imd appears to play a critical role in the tracheal system, while both Imd and Toll drive Defensin expression in the fat body (*Hoffmann and Reichhart, 2002*; *Tzou et al., 2002*). Consistently, overexpressing Defensin partially rescued the effect of Imd or Toll knockdown on *dlg* tumours. However, Toll and Imd pathways are major regulators of multiple AMPs in the *Drosophila* fat body. Then, it is conceivable that AMPs other than Defensin may possess similar anti-tumoural activity. In fact, a recent study in *Drosophila* shows that ectopic expression of several antimicrobial peptides, including Defensin, can increase cell death in a haematopoietic tumour model (*Araki et al., 2019*). Authors of that study also reported activation of Toll and Imd pathways together with increased expression of several AMPs in those tumour bearing animals. Together with our study, this suggests potentially general anti-tumoural properties of *Drosophila* AMPs.

## Defensin specifically targets tumour cells

Importantly, we show that Defensin targets tumour cells for apoptosis while sparing normal cells. As their human counterparts, we show that *Drosophila* tumours can expose PS independently of cell death (*Riedl et al., 2011*). Interestingly, an in vitro study showed selective anticancer activity of some synthetic peptides derived from beetle Defensin, against cancer cells exposing PS (*Iwasaki et al., 2009*). A similar targeting mechanism has also been proposed to explain temporin-1CEa or L-K6 anticancer activities towards melanoma cells and breast cancer cells respectively (*Wang et al., 2016*; *Wang et al., 2017*). PS is a mark of apoptotic cells, which is used as an 'eat me' signal by phagocytes (*Shklyar et al., 2013*; *Tung et al., 2013*). In *Drosophila*, the phagocytic receptor Simu together with Draper contribute to the elimination of apoptotic cells through recognition of PS (*Shklyar et al., 2013*; *Tung et al., 2013*). It would be interesting to test the role of these receptors in the control of tumour progression in *Drosophila*.

Our results indicate that PS exposure precedes Defensin action and is not just an 'eat-me' signal but likely contributes to changing the membrane of tumour cells making them sensitive to the action of AMPs. It is tempting to speculate that the deleterious effect of AMPs observed upon ageing or neurodegeneration may involve a similar targeting mechanism of 'foreign-looking' or unfit cells (*Cao et al., 2013*; *Kounatidis et al., 2017*; *Lezi et al., 2018*).

Noteworthy, other negatively charged molecules enriched on tumour surface such as heparan sulfates and O-glycosylated mucins may also contribute to the targeting by AMPs (*Hollingsworth and Swanson, 2004*; *Knelson et al., 2014*).

## Defensin action requires TNF

TNF is a major player in the tumour microenvironment by exerting pleiotropic effects on both the tumour and its surroundings (*Ham et al., 2016*; *Parameswaran and Patial, 2010*). In our *dlg* mutant model, we observed that defensin induction requires TNF, supporting previous observations that induction of innate immune response in tumour bearing animal relied on Egr produced by the tumour (*Parisi et al., 2014*). Indeed, Toll activation in the fat body has been proposed to be a consequence of tumour-derived TNF, which drives haemocyte proliferation leading to an increase in Toll ligand Spatzle (*Parisi et al., 2014*). Whether Imd pathway is also indirectly activated by the changes in immune cell activity in response to tumour or by alternative mechanisms remains an open question.

Our results show that haemocyte-derived TNF is required for PS exposure by the tumour, a key process for tumour targeting by Defensin. Consistently, haemocyte-derived TNF is shown to drive cell death in *dlg* mutant tumours (*Parisi et al., 2014*). The precise molecular mechanisms driving PS exposure downstream of TNF remain unknown. However, haemocyte-derived TNF, activates JNK pathway, which triggers many changes in tumour cells including apoptosis (*Igaki et al., 2002*; *Moreno et al., 2002*), ROS production (*Fogarty et al., 2016*; *Santabárbara-Ruiz et al., 2015*), loss of cell polarity (*Zhu et al., 2010*), modification of extracellular matrix (*Uhlirova and Bohmann, 2006*), proliferation and cell migration (*Beaucher et al., 2007*; *Igaki et al., 2006*; *Pastor-Pareja et al., 2004*; *Srivastava et al., 2007*). Mild activation of JNK on its own is insufficient to drive PS exposure and tissue sensitivity to Defensin-induced cell death (data not shown). While a JNK-independent role of Egr in PS exposure and sensitivity to the AMP cannot be ruled out, TNF might also sensitise cells through activation of the JNK pathway (*Cordero et al., 2010*) and PS exposure, thus providing a secondary sensitisation mechanism of tumour cells to the action of Defensin. Further studies are needed to explore a potential link between JNK activation and PS exposure.

Another consequence of TNF-dependent JNK activation in tumours is the increased expression of matrix metalloproteases (Mmps) by tumour cells (*Uhlirova and Bohmann, 2006*). Importantly, we show that Mmp-1 induction and then JNK pathway activation are still present in tumour from larvae devoided of Defensin. This demonstrates that TNF signalling acts upstream of Defensin. Mmps degrade the basal membrane facilitating metastasis of primary tumour cells (*Beaucher et al., 2007*; *Pastor-Pareja et al., 2004*; *Srivastava et al., 2007*). Interestingly, a study reported that human Mmp-7 can cleave immature Defensins, promoting their activation (*Wilson et al., 1999*; *Wilson et al., 2009*). Indeed, the pro-domain present in AMPs is thought to keep their in vivo activity silent, allowing local activation of AMP upon cleavage. While the mechanisms of *Drosophila*

Defensin pro-domain cleavage remain unknown, it would be interesting to explore whether the changes in the tumour microenvironment could also affect Defensin activation.

In conclusion, our study provides the first direct in vivo demonstration of the role of an endogenous AMP as an anti-cancer agent in *Drosophila*. Our data point to a conserved mechanism of tumour control by AMPs, a potent arm of the innate immune system. Importantly, we identify cellular features within tumours, which may be predictive of their sensitivity to be targeted by AMPs. This study provides a new paradigm to decipher the molecular mechanisms influencing anti-tumoural functions of an AMP, which may extend to other non-canonical roles of AMPs, such as in ageing, long-term memory and wound healing. Moreover, together with the new genetic tools allowing targeting of all *Drosophila* AMPs (*Hanson et al., 2019*), our study establishes new bases to explore in vivo a potential important natural mechanism of defence against tumours.

# Materials and methods

## Key resources table

| Reagent type (species) or resource | Designation | Source or reference | Identifiers | Additional information |
|---|---|---|---|---|
| Genetic reagent (*Drosophila melanogaster*) | w[1118] | (*Dewey et al., 2004*) | BDSC: 3605; RRID:BDSC_3605 | |
| Genetic reagent (*Drosophila melanogaster*) | w[1118] iso | (*Ferreira et al., 2014*) | N/A | |
| Genetic reagent (*Drosophila melanogaster*) | dlg[40.2]/FM7 | (*Mendoza-Topaz et al., 2008*) | Flybase_FBal0240608 | |
| Genetic reagent (*Drosophila melanogaster*) | FRT82B,scrib[1]/TM6 | (*Bilder et al., 2000*) | Flybase_FBal0103577 | |
| Genetic reagent (*Drosophila melanogaster*) | egr[3] | (*Igaki et al., 2002*) | Flybase_FBal0147163 | |
| Genetic reagent (*Drosophila melanogaster*) | imd[1] | (*Leulier et al., 2000*) | Flybase_FBal0045906 | |
| Genetic reagent (*Drosophila melanogaster*) | rel[E20] | (*Leulier et al., 2000*) | DGGR: 109927; RRID:DGGR_109927 | |
| Genetic reagent (*Drosophila melanogaster*) | def[sk3] | (*Hanson et al., 2019*) | N/A | |
| Genetic reagent (*Drosophila melanogaster*) | def[sk4] | (*Hanson et al., 2019*) | N/A | |
| Genetic reagent (*Drosophila melanogaster*) | btl-gal4,UAS-RFP/CyO | Irene Miguel-Aliaga | N/A | |

*Continued on next page*

*Continued*

| Reagent type (species) or resource | Designation | Source or reference | Identifiers | Additional information |
|---|---|---|---|---|
| Genetic reagent (*Drosophila melanogaster*) | lpp-gal4/TM6B | (*Brankatschk and Eaton, 2010*) | N/A | |
| Genetic reagent (*Drosophila melanogaster*) | tub-gal4 | Bloomington *Drosophila* Stock Center | BDSC: 5138; RRID:BDSC_5138 | y(1) w[*]; P{w[+mC]=tubP-GAL4}LL7/TM3, Sb(4) Ser(1) |
| Genetic reagent (*Drosophila melanogaster*) | hml$^\Delta$-gal4,UAS-gfp | Bruno Lemaitre | N/A | |
| Genetic reagent (*Drosophila melanogaster*) | en-gal4 | Bloomington *Drosophila* Stock Center | BDSC: 30564; RRID:BDSC_30564 | y1 w*; P{w + mW.hs=en2.4 GAL4}e16E |
| Genetic reagent (*Drosophila melanogaster*) | UAS-def IR | Vienna *Drosophila* Resource Centre | VDRC: 102437; RRID: Flybase_FBst0474306 | P{KK111656}VIE-260B |
| Genetic reagent (*Drosophila melanogaster*) | UAS-imd IR | Vienna *Drosophila* Resource Centre | VDRC: 101834; RRID: Flybase_FBst0473707 | P{KK109863}VIE-260B |
| Genetic reagent (*Drosophila melanogaster*) | UAS-myd88 IR | Vienna *Drosophila* Resource Centre | VDRC: 25402; RRID: Flybase_FBst0455868 | w$^{1118}$; P{GD9716}v25402 |
| Genetic reagent (*Drosophila melanogaster*) | UAS-dlg IR | Vienna *Drosophila* Resource Centre | VDRC: 41136; RRID: Flybase_FBst0463952 | w$^{1118}$; P{GD4689}v41136/TM3 |
| Genetic reagent (*Drosophila melanogaster*) | UAS-egr IR | Vienna *Drosophila* Resource Centre | VDRC: 108814; RRID: Flybase_FBst0480608 | P{KK103432}VIE-260B |
| Genetic reagent (*Drosophila melanogaster*) | UAS-w IR | Bloomington *Drosophila* Stock Center | BDSC: 25785; RRID:BDSC_25785 | y(1) v(1); P{y[+t7.7] v[+t1.8]=TRiP.JF01786}attP2 |
| Genetic reagent (*Drosophila melanogaster*) | UAS-def | (*Tzou et al., 2000*) | Flybase_FBal0145092 | |
| Genetic reagent (*Drosophila melanogaster*) | UAS-def-3xHA | FlyORF | FlyORF: F002467; RRID:Flybase_FBal0298643 | M{UAS-Def.ORF.3xHA.GW}ZH-86Fb |
| Genetic reagent (*Drosophila melanogaster*) | UAS-dcr2 | Bloomington *Drosophila* Stock Center | BDSC: 24650; RRID:BDSC_24650 | w[1118]; P{w[+mC]=UAS-Dcr-2.D}2 |
| Genetic reagent (*Drosophila melanogaster*) | spz$^{M7}$ | (*Neyen et al., 2014*) | N/A | |
| Antibody | Anti-GFP (Chicken polyclonal) | Abcam | Cat# ab13970; RRID:AB_300798 | IF(1:4000) |

*Continued on next page*

*Continued*

| Reagent type (species) or resource | Designation | Source or reference | Identifiers | Additional information |
|---|---|---|---|---|
| Antibody | Anti-Def (Mouse polyclonal) | Dahua Chen (*Ji et al., 2014*) | N/A | IF(1:100) |
| Antibody | Anti-HA (Mouse monoclonal) | Cell Signaling Technology | Cat# 2367, RRID:AB_10691311 | IF(1:1000) |
| Antibody | Anti-dcp1 (Rabbit polyclonal) | Cell Signaling Technology | Cat# 9578, RRID:AB_2721060 | IF(1:100) |
| Antibody | Anti-phospho-Histone H3 (Ser10) (Rabbit polyclonal) | Cell Signaling Technology | Cat# 9701, RRID:AB_331535 | IF(1:100) |
| Antibody | Anti-phospho-Histone H3 (Ser28) (Rabbit polyclonal) | Cell Signaling Technology | Cat# 9713, RRID:AB_823532 | IF(1:100) |
| Antibody | Anti-Mmp1 (Mouse clonality unknown) | Developmental Studies Hybridoma Bank | Cat# 3B8D12, RRID:AB_579781 | IF(1:10) |
| Antibody | Anti-Chicken IgY Alexa 488 (Goat polyclonal) | Molecular Probes | Cat# A-11039, RRID:AB_142924 | IF(1:500) |
| Antibody | Anti-Mouse IgG Alexa 488 (Goat polyclonal) | Molecular Probes | Cat# A-11029, RRID:AB_138404 | IF(1:500) |
| Antibody | Anti-Mouse IgG Alexa 594 (Goat polyclonal) | Molecular Probes | Cat# A-11032, RRID:AB_141672 | IF(1:500) |
| Antibody | Anti-Rabbit IgG Alexa 488 (Goat polyclonal) | Molecular Probes | Cat# A-11008, RRID:AB_143165 | IF(1:500) |
| Antibody | Anti-Rabbit IgG Alexa 594 (Goat polyclonal) | Thermo Fischer Scientific | Cat# A-11037, RRID:AB_2534095 | IF(1:500) |
| Sequence-based reagent | rpl32-fwd | This paper | PCR primers | AGGCCCAAGATCGTGAAGAA |
| Sequence-based reagent | rpl32-rev | This paper | PCR primers | TGTGCACCAGGAACTTCTTGA |
| Sequence-based reagent | def-fwd | This paper | PCR primers | CTTCGTTCTCGTGGCTATCG |
| Sequence-based reagent | def-rev | This paper | PCR primers | ATCCTCATGCACCAGGACAT |
| Sequence-based reagent | def-PCR-fwd | This paper | PCR primers | TTATTGCAGAAACGGGCTCT |
| Sequence-based reagent | def-PCR-rev | This paper | PCR primers | ATGGTAAGTCGCTAACGCTAATG |
| Sequence-based reagent | def-seq | This paper | Sequencing primers | CGTGTCTTCCTGCACAGAAA |
| Sequence-based reagent | attA-fwd | This paper | PCR primers | ATGCTCGTTTGGATCTGACC |
| Sequence-based reagent | attA-rev | This paper | PCR primers | TCAAAGAGGCACCATGACCAG |
| Sequence-based reagent | cecA1-fwd | This paper | PCR primers | CTCAGACCTCACTGCAATAT |
| Sequence-based reagent | cecA1-rev | This paper | PCR primers | CCAACGCGTTCGATTTTCTT |

*Continued on next page*

*Continued*

| Reagent type (species) or resource | Designation | Source or reference | Identifiers | Additional information |
|---|---|---|---|---|
| Sequence-based reagent | dro-fwd | This paper | PCR primers | CGTTTTCCTGCTGCTTGCTT |
| Sequence-based reagent | dro-rev | This paper | PCR primers | GGCAGCTTGAGTCAGGTGAT |
| Sequence-based reagent | drs-fwd | This paper | PCR primers | CTCTTCGCTGTCCTGATGCT |
| Sequence-based reagent | drs-rev | This paper | PCR primers | ACAGGTCTCGTTGTCCCAGA |
| Peptide, recombinant protein | *Drosophila* endogenous Defensin | Bulet EIRL | N/A | |
| Peptide, recombinant protein | *Drosophila* Synthetic Defensin | Genepep | N/A | ATCDLLSKWNWNHTACAGH CIAKGFKGGYCNDKAVCVCRN |
| Commercial assay or kit | High Capacity cDNA Reverse Transcription Kit | Applied Biosystems | Cat# 4368813 | |
| Commercial assay or kit | PerfeCTa SYBR Green FastMix (Low ROX) | Quanta Bio | Cat# 95074–012 | |
| Commercial assay or kit | TRIzol Reagent | Thermo Fisher Scientific | Cat# 15596018 | |
| Commercial assay or kit | Turbo DNA free Kit | Life Technologies LTD | Cat# AM1907 | |
| Commercial assay or kit | High Capacity cDNA Reverse Transcription Kit | Applied Biosystems | Cat# 4368813 | |
| Chemical compound, drug | 4′,6-Diamidine-2′-phenylindole dihydrochloride (DAPI) | Sigma | Cat# D9542 | 1 μg/mL |
| Software, algorithm | Fiji | NIH | https://fiji.sc/ | |
| Software, algorithm | GraphPad Prism 6 | GraphPad | RRID:SCR_002798 | |
| Software, algorithm | 7500 Real-Time PCR Software | Applied Biosystems | RRID:SCR_014596 | |
| Software, algorithm | BatchQuantify | (*Johansson et al., 2019*) | https://github.com/emltwc/2018-Cell-Stem-Cell | |
| Software, algorithm | GraphPad Prism 6 | GraphPad | RRID:SCR_002798 | |
| Software, algorithm | Volocity 3D Image Analysis Software | Perkin Elmer | RRID:SCR_002668 | |
| Software, algorithm | ZEN two lite | Zeiss | RRID:SCR_013672 | |
| Other | RNasine Plus RNase Inhibitor | Promega | Cat# N261 | |
| Other | Vectashield mounting media | Vector Laboratories, Inc. | Cat# H-1000–10 | |
| Other | Annexin V, Alexa Fluor 568 conjugate | Life Technologies LTD | Cat# A13202 | 1:20 |
| Other | Annexin V, Alexa Fluor 488 conjugate | Life Technologies LTD | Cat# A13201 | 1:20 |

*Continued on next page*

*Continued*

| Reagent type (species) or resource | Designation | Source or reference | Identifiers | Additional information |
|---|---|---|---|---|
| Other | Penicillin-Streptomycin (10,000 U/mL) | Thermo Fisher Scientific | Cat# 15140122 | Pen: 100 IU/mL Strep: 100 mg/mL |
| Other | PfuUltra II Fusion HS DNA Polymerase | Agilent | Cat# 600670 | |
| Other | AnnexinV binding buffer | Fisher Scientific | Cat# BDB556454 | |

## *Drosophila* housekeeping and rearing

Flies were reared at 25°C under 12 hr/12 hr light/dark cycles on standard oatmeal and molasses medium (*Fernandez-Ayala et al., 2009*). Details on fly strains used in this study are presented in the key resources table and detailed genotypes are included in corresponding figure legends.

For assessment of tumour volume and tumour cell death, flies were transferred to medium embryo collection cages and allowed to lay eggs for 8 hr at room temperature (RT) on agar/grape juice plates (2.1% agar, 25% grape juice, 1.25% sucrose, 0.2% methyl 4-hydroxybenzoate) supplemented with yeast paste. Plates were kept at 25°C until hatching. 200 larvae hatched within a 4 hr time window were collected and transferred to rearing bottles containing 45 mL of fly food to avoid any developmental delay and/or starvation effects due to overcrowding conditions. The tumour phenotype was analysed close to the normal developmental time, i.e. 7 days after hatching. Tumours were dissected and stained for further analysis (see below).

## Generation of *defensin* mutants

$def^{sk3}$ and $def^{sk4}$ mutants were generated using CRISPR/cas9 technology (*Hanson et al., 2019*). Mutants were isogenised through backcrosses with a $w^{1118}$ *iso* line as previously described in *Ferreira et al. (2014)*.

## Survival assays

Survival of $def^{SK3}$ animals was compared to wild type controls (isogenised background, *iso*). Male flies (5–7 days old) were pricked in the thorax with a needle dipped in a concentrated pellet of fresh overnight bacterial culture. Infected flies were kept at 29°C. *Listeria innocua* was cultured in Brain heart infusion medium at 37°C and used at optical density of 200 nm ($OD_{600}$). *Erwinia carotovora carotovora 15* was cultured in Luria-Bertani broth at 29°C and used at $OD_{600}$ 200. Experiments were repeated three times independently and one representative experiment is shown.

## Immunohistochemistry

Tissues were dissected in Phosphate Buffer Saline (PBS) and fixed for 20 min in 4% formaldehyde. Fat body samples were fixed for 30 min using the same protocol. Tissues were washed three times in PBS containing 1% of Triton-X100 (PBT) and incubated overnight at 4°C with primary antibodies. Tissues were washed five times in PBT and incubated for 2 hr at RT with secondary antibodies and 4′,6-Diamidine-2′-phenylindole dihydrochloride (DAPI). After three washes in PBT, tissues were mounted in Vectashield using Secured-Seal spacers (Thermo Fisher Scientific).

Confocal images were captured using a Zeiss 710 or Zeiss 880 with Airyscan confocal microscope and processed with Fiji 2.0.0 or Adobe Photoshop C.S5.1.

## Quantification of tumour volume and tumour cell death

For analysing tumour phenotype, images were acquired using optimal slice parameter and 12bits using a Zeiss 710 confocal microscope. Quantifications were made as previously described (*Parisi et al., 2014*). Briefly, we used Volocity 3D imaging analysis software to quantify the total volume of tumour identified by DAPI staining and cell death visualised with anti-Dcp1 staining. When haltere or leg discs tumours were still associated with the wing disc, they were not considered for quantification. Quantifications were done in at least three independent biological replicates for each genetic setting. Single representative experiments are presented except for *dlg;def* double mutant

rescue experiments (*Figure 2K–L*) for which replicates are pooled due to the few larvae available. In all cases the difference between control and mutant tumour were strongly reproducible between replicates.

## Quantification of colocalisation between Def and Dcp1 or Annexin V

To quantify Def/Dcp-1 and Def/Annexin V colocalisation, we determined the intensity of Def staining in areas of positive versus negative staining for the second marker using the ImageJ macro, Batch-Quantify (*Johansson et al., 2019*).

## Sterile rearing conditions

To rule out any effect of infection on the tumour phenotypes, *Drosophila* larvae were reared on standard food supplemented with penicillin and streptomycin when indicated in the figure.

## qRT-PCR analysis

Total RNA from 7 to 10 whole larvae or tissues collected from 10 to 20 larvae (10 larvae for fat body, 20 larvae for trachea) was extracted using TRIzol according to the manufacturer's instructions. RNA was treated with Turbo DNA free Kit and RNA concentration and quality were monitored using a Nanodrop (Thermo Fisher Scientific). The same amount of total RNA (1 µg for whole larvae and fat body or 100 ng for trachea) was used to perform the three independent reverse-transcriptions using the High-Capacity cDNA Reverse Transcription Kit. cDNAs were pooled and qPCR was performed using PerfeCTa SYBR green following the manufacturer's instructions. cDNA amplification was monitored with Applied Biosystems 7500 fast instruments. Serial 10-fold dilutions of an external standard were used to produce a standard curve, and RNA samples were included to control for the absence of DNA contamination. *rpl32* expression was used to normalise expression levels of target genes. Data was analysed using the Ct method ($2^{-\Delta\Delta ct}$). All qPCR experiments were carried out on three to seven independent biological replicates (see figure legends for details) and, larvae were sampled from the same bottle used to analyse the tumour phenotype. GraphPad Prism7 software was used for graphical representation and statistical analysis. Primer targets and sequences are presented in key resources table.

## PCR amplification of genomic DNA

To verify the sequence of *def^{sk3}* and *def^{sk4}* mutants, genomic DNA was extracted from single flies by grinding whole animals with a 200 µL pipette tip containing 50 µL of squishing buffer (10 mM Tris-HCl pH 8.2, 1 mM EDTA, 25 mM NaCl, 200 µg proteinase K) followed by incubation at 37°C for 30 min. Proteinase K was heat-inactivated at 95°C for 2 min. 1 µL of genomic DNA was used for PCR amplification using PfuUltra II Fusion DNA Polymerase and Eppendorf Mastercycler Ep Gradient Thermal Cycler.

## Annexin V staining and quantification

Tumours were dissected in PBS and incubated in Annexin V binding buffer containing 5% of Alexa Fluor (488 nm or 568 nm) conjugated Annexin V for 10 min. Tissues were washed quickly in Annexin V binding buffer and fixed in 4% formaldehyde for 20 min. Immunostaining, mounting and imaging were then carried out as described above.

We used Fiji 2.0.0 software to quantify Annexin V staining at the tumour surface. Maximal Z-projection of whole tumours were generated and the Colour Threshold tool was then used to detect the red pixels that represented Annexin V staining at the surface of tumours. The hue slider was set to include only the red signal, and the brightness slider was adjusted to exclude any background signal. This area was then selected and measured in pixels.

## Synthetic defensin production and injections

The mature Defensin peptide was synthesised (over 90% purity) and the purity controlled by Gene-pep using Ultra Performance Liquid Chromatography/Mass Spectrometry. We subsequently compared the integrity of synthetic Defensin with authentic Defensin purified from *Drosophila* hemolymph using two complementary approaches, namely Matrix-Assisted Laser Desorption/Ionisation-Mass Spectrometry (MALDI-MS) for molecular mass determination and trypsin digestion

followed by nanoLiquid Chromatography/Electrospray Ionisation tandem Mass Spectroscopy (nano*LC/ESI MS/MS*) for molecular integrity confirmation. Those analyses confirmed the similar mass between natural and synthetic Defensin (respectively provided by Bulet EIRL and Genepep) as well the presence of disulphide bonds within synthetic Def, demonstrating it is the mature peptide (Data not shown).

For injection experiments, larvae were collected, washed three times in cold sterile PBS and injected on a fly pad upon $CO_2$ anaesthesia using Nanoject II (Drummond) and glass capillaries 3.5 inch (Drummond). Larvae were injected with 69 nL of PBS containing 7.5 nmol of synthetic Defensin or 69 nL of PBS only as control. Larvae were then gently transferred into agar/grape juice plates and kept at 25°C. Tumours were dissected 4 hr after injection and stained for visualisation of nuclei and tumour cell death as describe above.

### Statistical analyses

All data are presented as mean ± SD and n values are indicated in the figure legends. Statistical analyses were carried out using GraphPad Prism7 software and only significant differences are indicated in dot-plots. Survival upon infection was analysed using Log-rank test. qPCR data were analysed using unpaired t-test with a two-tailed p value. Tumour volume and tumour cell death comparing two samples were analysed using Mann-Whitney test with a two-tailed p value and the one comparing three genotypes were analysed using One-way ANOVA followed by Turkey's multiple comparisons test. Tumour volume from injected larvae was analysed using Two-way ANOVA followed by Turkey's multiple comparisons test and only significant statistical differences between ctrl and Defensin injected larvae for each genotype tested are indicated.

### Data and material availability

All raw data is available through http://dx.doi.org/10.5525/gla.researchdata.834.

## Acknowledgements

We would like to thank the Vienna *Drosophila* Resource Centre, the Bloomington *Drosophila* Stock Centre and Mirka Uhlirova for fly stocks. We are thankful to Core Services and Advanced Technologies at the Cancer Research UK Beatson Institute (C596/A17196), with particular thanks to Beatson Advanced Imaging Resource. We are very grateful to Dr. Dahua Chen for sharing the anti-Defensin antibody, to Dr. Sébastien Voisin for his critical contribution in mass spectrometry analyses and to Dr. Mate Naszai for critical help in the quantification of staining co-localization. JPP and YY are supported by Cancer Research UK core funding through the CRUK Beatson Institute (A17196). JBC is a Sir Henry Dale Fellow jointly funded by the Wellcome Trust and the Royal Society (grant number 104103/Z/14/Z). *In memory of Marcos Vidal (1974–2016).*

## Additional information

#### Competing interests

Bruno Lemaître: Reviewing Editor, *eLife*. The other authors declare that no competing interests exist.

#### Funding

| Funder | Grant reference number | Author |
|---|---|---|
| Cancer Research UK | A17196 | Julia B Cordero |
| Wellcome | 104103/Z/14/Z | Julia B Cordero |

The funders had no role in study design, data collection and interpretation, or the decision to submit the work for publication.

## Author contributions
Jean-Philippe Parvy, Conceptualization, Data curation, Formal analysis, Validation, Investigation, Visualization, Methodology, Writing—original draft, Writing—review and editing, Performed most of the experiments in this study; Yachuan Yu, Alina Kurjan, Formal analysis, Investigation, Provided technical assistance; Anna Dostalova, Formal analysis, Investigation, Assayed survival upon infection in Def mutant flies (Figure 1-figure supplement 1); Shu Kondo, Investigation, Generated the Defensin mutant flies; Philippe Bulet, Resources, Formal analysis, Investigation, Writing—original draft, Analysed the quality of synthetic defensin; Bruno Lemaître, Resources, Formal analysis, Writing—original draft, Generated the defensin mutant flies, Provided useful comments and advice on the original manuscript draft; Marcos Vidal, Conceptualization, Initiated project; Julia B Cordero, Conceptualization, Data curation, Formal analysis, Supervision, Funding acquisition, Validation, Methodology, Writing—original draft, Project administration, Writing—review and editing, Directed the study

## Author ORCIDs
Jean-Philippe Parvy (ID) https://orcid.org/0000-0002-6643-5686
Bruno Lemaître (ID) https://orcid.org/0000-0001-7970-1667
Julia B Cordero (ID) https://orcid.org/0000-0003-1701-9480

## Decision letter and Author response
Decision letter https://doi.org/10.7554/eLife.45061.018
Author response https://doi.org/10.7554/eLife.45061.019

## Additional files

### Supplementary files
• Transparent reporting form
DOI: https://doi.org/10.7554/eLife.45061.016

### Data availability
All raw data are available through http://dx.doi.org/10.5525/gla.researchdata.834. Owing to the size of the dataset, it can be requested through the link.

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
