## [Decision Letter]

Thank you for submitting your article "The antimicrobial peptide Defensin cooperates with Tumour Necrosis Factor to drive tumour cell death in *Drosophila*" for consideration by *eLife*. Your article has been reviewed by two peer reviewers, and the evaluation has been overseen by a Reviewing Editor and Utpal Banerjee as the Senior Editor. The following individual involved in review of your submission has agreed to reveal their identity: Georg Halder (Reviewer #3).

The reviewers have discussed the reviews with one another and the Reviewing Editor has drafted this decision to help you prepare a revised submission.

This is an interesting manuscript about a topic of wide significance. The authors investigate the role of the antimicrobial peptide defensin in killing tumour cells in a *Drosophila* cancer model. They report that defensin is secreted by the fat body and trachea in response to Toll and Imd signalling and that defensin kills tumour cells by binding to cell surface phosphatidylserine, which become exposed in response to TNF signalling. This is novel and, if true, may provide the foundation of new approaches to understanding cell death and growth control in tumours.

For a rather radical proposal, the data need to be solid. In general, the results are convincing and carefully controlled but here are a few that are very important to the model but are not as convincing (or at least not as clearly presented) as most of the others. The reviewers' consensus is that the following issues need to be addressed before this paper could be accepted by *eLife*.

1) In Figure 4E they claim to show that humoral defensin is binding to 'transformed imaginal discs' in *dlg* mutant animals. This figure is hard to interpret. Tumour tissue seems to be marked by anti-Dcp1, a cell death marker, but it's not clear whether this marks the whole tumour. Moreover, the HA-defensin staining is very limited and does not obviously colocalise with Dcp1 or anything else. The idea that defensin binds to tumour tissue is central to the model 4E does not strongly support the idea. To be fair, the other data in Figure 4 are clearer and more convincing, but 4E is a problem. It would be better to show an image of the whole disc first so that we can appreciate the morphology and also how the staining looks over an entire tumour and not only in a small selected region. It would also be good to show the individual channels of 4G and to comment on the amount of non-overlap that is present.

2) Aspects of Figure 7, which is central to the conclusion that defensin binds to exposed PS on tumour cells, are also of concern. Panels A purport to show that defensin colocalises with annexin V, a marker of exposed PS, but the images do not show that convincingly. It is clear in A' and A' that there is plenty of non-overlap between the green annexin and red defensin. Much red defensin appears to label annexin-negative tissue. Without clearer data, preferably quantified, it is hard to support the conclusion.

3) The model that JNK activation is a prerequisite for defensin mediated tumour cell killing is of particular interest, as AMPs lack strong specificity towards killing cancer cells. However, since the experiments were done in *egr* mutants, which is a secreted protein that activates JNK signalling, it is not clear whether the requirement for JNK activation is in tumour cells and/or other cells. JNK could just as well act non-autonomously and this experiment would distinguish whether their model is correct or not. To further investigate this, the authors should test the effect of defensin overexpression on the efficacy of JNK induced cell death. This might best be done in clones with weak JNK activation, e.g. puckered hypomorphic clones. However, a scenario of double overexpression of defensin with a construct to activate JNK may be sufficient.

4) It is notable that the data in Figure 7C shows that the statistical confidence of the differences is rather low.

5) Both the written description and the diagrams in Figures 1 and 7 of the complex signalling between tumour cells, haemocytes, FB and trachea, involving TNF, Toll, Imd, and defensin, are complex and hard to follow. Of course this is partly because biology is complicated, but a better job could be done. It is important that readers do understand both the anatomical and the signalling aspects of the model.

---

## [Author Response]

[…] For a rather radical proposal, the data need to be solid. In general, the results are convincing and carefully controlled but here are a few that are very important to the model but are not as convincing (or at least not as clearly presented) as most of the others. The reviewers' consensus is that the following issues need to be addressed before this paper could be accepted by eLife.1) In Figure 4E they claim to show that humoral defensin is binding to 'transformed imaginal discs' in dlg mutant animals. This figure is hard to interpret. Tumour tissue seems to be marked by anti-Dcp1, a cell death marker, but it's not clear whether this marks the whole tumour. Moreover, the HA-defensin staining is very limited and does not obviously colocalise with Dcp1 or anything else. The idea that defensin binds to tumour tissue is central to the model 4E does not strongly support the idea. To be fair, the other data in Figure 4 are clearer and more convincing, but 4E is a problem. It would be better to show an image of the whole disc first so that we can appreciate the morphology and also how the staining looks over an entire tumour and not only in a small selected region. It would also be good to show the individual channels of 4G and to comment on the amount of non-overlap that is present.

The goal of Figure 4E was to unambiguously show that none-tumour produced Def (fat body overexpressed Def-HA) binds to tumours but not wild type tissues (old Figure 4F, F’). As the reviewer pointed out, the assessment of endogenous Def in tumours does supports the selective binding of the AMP to the tumours (new Figure 4D-H’’). However, the reviewer is correct in that old Figure 4E was not clear enough to make the case for Def tumour binding and/or Dcp-1 co-localization. One of the issues is that, in addition to the tumour itself, tumour associated tissues, which are negative for Dcp-1 but produce Defensin (fat body and trachea) are almost always captured in these images. Def expression in such accessory tissues is obviously more robust in the context of Def-HA overexpression (Figure 4—figure supplement 1). We now provide new and significantly improved images of a whole tumour and zoomed-in areas of tumour tissue, including individual channels (new Figure 4A-B’’). For completeness, we also present quantifications, which confirm enrichment of endogenous Def in Dcp-1^+ve^ areas on the tumours (new Figure 4F). For organisational purposes, pictures depicting basal UAS-Def-HA expression patterns and following fat body overexpression in wild type discs (old Figure 4A-D) are now in a new Figure 4—figure supplement 1.

Furthermore to unambiguously demonstrate the specificity of ectopically produced Def to bind to tumours exposing PS, we overexpressed Def-HA in the fat body of *dlg^40.2^;egr^3/3^* animals (new Figure 7—figure supplement 1A, B). *dlg^40.2^;egr^3/3^* tumours don’t expose PS and are refractive to Def killing action (Figures 7G and K). Consistently, they present no detectable staining for Def-HA (new Figure 7—figure supplement 1C, C’).

2) Aspects of Figure 7, which is central to the conclusion that defensin binds to exposed PS on tumour cells, are also of concern. Panels A purport to show that defensin colocalises with annexin V, a marker of exposed PS, but the images do not show that convincingly. It is clear in A' and A' that there is plenty of non-overlap between the green annexin and red defensin. Much red defensin appears to label annexin-negative tissue. Without clearer data, preferably quantified, it is hard to support the conclusion.

As for the previous point we now present quantification of Def and AnnexinVco-localization (new Figure 7C). We also present individual channels for a more clear visualization of Annexin V and Def staining (new Figure 7A-A’’ and B-B’’). As noted by the reviewer, there are tumour areas stained with Def and not AnnexinV and vice versa. However, our data from images and quantifications clearly shows an enrichment of Def within AnnexinV^+ve^ versus AnnexinV^-ve^ tumour areas. Moreover, a new experiment showing that Def-HA can’t associate to *dlg^40.2^;egr^3/3^* tumours also support this conclusion (see above and new Figure 7—figure supplement 1C, C’).

3) The model that JNK activation is a prerequisite for defensin mediated tumour cell killing is of particular interest, as AMPs lack strong specificity towards killing cancer cells. However, since the experiments were done in egr mutants, which is a secreted protein that activates JNK signalling, it is not clear whether the requirement for JNK activation is in tumour cells and/or other cells. JNK could just as well act non-autonomously and this experiment would distinguish whether their model is correct or not. To further investigate this, the authors should test the effect of defensin overexpression on the efficacy of JNK induced cell death. This might best be done in clones with weak JNK activation, e.g. puckered hypomorphic clones. However, a scenario of double overexpression of defensin with a construct to activate JNK may be sufficient.

This is indeed a very interesting question. However, we would like to point that, our data definitely suggests that Egr and PS exposure, which is dependent on Egr, are a pre-requisite for Def killing of tumour cells. Even though Egr does activate JNK signaling, it may also be doing more than that in these tumours. In order to best address the reviewer’s question following his/her line of thought and with the tools that were available to us to perform experiments within the timeframe of this revision we did the following:

1- To achieve mild JNK activation as we successfully did in other contexts (Cordero et al., 2012), we overexpressed wild type JNK. Here, we used *engrailed-gal4 (eng>hep^wt^*) and analysed wing discs. This condition, led to mild apoptosis in the wing pouch but was not sufficient to drive detectable PS exposure (Author response image 1). Therefore, and consistently with our model (PS exposure is required for Def killing), the administration of injected recombinant Def or genetic co-expression of UAS-Def did not modify JNK-dependent cell death (Author response image 1).

2- Given the mild basal phenotype obtained by overexpressing wild type JNK and following the second suggestion by the reviewer “a scenario of double overexpression of defensin with a construct to activate JNK may be sufficient*”.* We overexpressed activated *hemipterous* using *rotund-gal4 (rn>hep^act^*) as *eng>hep^act^* led to embryonic lethality. However, *rn>hep^act^*discs presented a complete apoptosis of the wing pouch, and consistently extremely rare adult escapers were devoid of wings (Author response image 2 and data not shown). We reasoned that a quantifiable assessment of a potential enhancement of this phenotype was not going to be possible.

As per the results with *hep^wt^*, we can conclude that a mild activation of JNK in an otherwise wild type tissue is not sufficient to allow apoptosis by Defensin. Nonetheless, we cannot exclude that an intermediate, sublethal, activation of the JNK pathway might lead to Def sensitivity. Alternatively, and given the absolute need of Egr for PS exposure for the Def anti-tumoural role, it is conceivable that the role of Egr in these tumours extends beyond activation of JNK. We have integrated these results in the revised Discussion of our manuscript.

**Author response image 1. respfig1:** Anti-tumoural immunity in flies: the key players. (**A, B**) Third instar larval imaginal wing disc overexpressing of *hep^wt^* using *engrailed-gal4, UAS-rfp (eng>rfp; hep^wt^*) (**A**; red). Discs where stained with AnnexinV to visualize PS exposure (**B**; green). (**C**) Assessment of apoptotic cell death in wing discs from *eng>rfp; hep^wt^*animals injected with PBS (control) or recombinant Defensin. (**D**) Assessment of apoptotic cell death in wing discs from animal following overexpression of *hep^wt^*only, *def* only, or co-expression of both transgenes using combined *eng-gal4* and *lpp-gal4* drivers.

**Author response image 2. respfig2:** Anti-tumoural immunity in flies: the key players. (A, B) Third instar larval imaginal wing disc overexpressing of *hep^act^* using *rutond-gal4 (rn>hep^act^*). Discs where stained with anti-Dcp1 to visualize apoptotic cell death (B; red).

4) It is notable that the data in Figure 7C shows that the statistical confidence of the differences is rather low.

This is a fair point. Reasons for this are likely due to the variability in *def* expression levels within the *dlg* animals. Moreover, we have done ANOVA to compare multiple samples in old Figure 7C (new Figure 7D), which is more stringent than T-test used in other cases in the manuscript where we compared only two sets of RT-QPCR data (e.g. Figure 1B). Therefore, even * is meaningful. Importantly, we do present several lines of evidence on the functional importance of Egr in the system.

5) Both the written description and the diagrams in Figures 1 and 7 of the complex signalling between tumour cells, haemocytes, FB and trachea, involving TNF, Toll, Imd, and defensin, are complex and hard to follow. Of course this is partly because biology is complicated, but a better job could be done. It is important that readers do understand both the anatomical and the signalling aspects of the model.

We have now simplified the diagrams (new Figure 1A and Figure 7L) and text to focus on the key points addressed in this manuscript.